# A revised ocean glider concept to realize Stommel's vision and supplement Argo floats

Erik M. Bruvik[1], Ilker Fer[1,2], Kjetil Våge[1,2], Peter M. Haugan[1,3]

[1]Geophysical Institute, University of Bergen, Bergen, Norway
[2] Bjerknes Centre for Climate Research, University of Bergen, Bergen, Norway
[3] Institute of Marine Research, Bergen, Norway

*Correspondence to*: Ilker Fer (ilker.fer@uib.no)

**Abstract.** This paper revisits Stommel's vision for a global glider network and the Argo design specification. A concept of floats with wings, so-called slow underwater gliders, is explored. An analysis of the energy/power consumption shows that,
by operating gliders with half the vehicle volume at half the speed compared to present gliders, the energy requirements for long duration missions can be met with available battery capacities. Simulation experiments of slow gliders are conducted using the horizontal current fields from an eddy-permitting ocean reanalysis product. By employing a semi-Lagrangian, streamwise navigation whereby the glider steers at right angles to ocean currents, we show that the concept is feasible. The simulated glider tracks demonstrate the potential for efficient coverage of key oceanographic features and variability.

**1 Introduction**

In Stommel's (1989) vision for the year 2021, oceans would be monitored using instruments with wings. These robots would profile through the water column by changing their buoyancy in alternating vertical cycles of ascents and descents, with their wings providing the horizontal propulsion to "glide" through the oceans. It is now timely to look back and revisit this vision and assess status.

Today, the oceans are indeed extensively monitored by buoyancy driven robots – albeit without wings. These instruments are called floats, and approximately 4000 floats profile the oceans in the Argo programme (Roemmich et al., 2009). Their contribution to the knowledge of the oceans is substantial (Riser et al., 2016), yet, juxtaposing the Argo-programme and Stommel's vision invites a curious investigation as to why the floats lack wings. In that sense floats fall short of realizing his
vision recognizing that wings would also allow for dynamic positioning. The original Argo design specification (Roemmich et al., 1999, p.3) explicitly mentions the possibility of a winged gliding float:

> "…, a profiling float equipped with wings for dynamic positioning during ascent and descent, offers further potential. This "gliding" float will provide a similar number of T/S [temperature and salinity] profiles at a fixed location or
along a programmed track."

The stated potential has not yet been realized, and further motivates the inquiry presented here.

Floats, without wings, are now a robust and mature technology developed since the 1950's (Gould, 2005; Davis et al., 2001). Underwater gliders, floats with wings (hereafter referred to as gliders), are a newer, more complex development starting in the 1990's. The first successful glider designs materialized in the early 2000s (Davis, Eriksen and Jones, 2002; Jenkins et al., 2003), and have by now demonstrated their role as a reliable and useful tool for oceanographic exploration of phenomena, e.g. boundary currents where floats typically have short residence times (Rudnick, 2016; Lee and Rudnick, 2018). As the glider technology matured, the enthusiasm has gradually cooled (Rudnick, 2016). Gliders also fell short of realizing Stommel's vision. For instance, Stommel envisioned a global glider effort, compared to the current regional efforts; he envisioned endurance of years compared to months; and he foresaw 1000 gliders compared to a few tens in operation simultaneously. In one aspect current gliders do meet Stommel's expectation: their horizontal velocity is indeed approximately 25 cm s[-1]. We will, in the following, dispense with this latter requirement and argue that his vision may thereby be potentially realized.

Stommel never seriously assessed the power requirements, suggesting instead that gliders could harvest energy from the ocean thermocline. This has proved less than practical and is also not a solution for the global ocean.

Current glider designs (Sherman et al., 2001; Eriksen et al., 2001; and Webb et al., 2001) operate roughly according to the maxim "1/2 knot at 1/2 Watt", i.e. they glide through the ocean at a horizontal velocity of roughly half a knot (25 cm s[-1]) consuming about half a Watt of power. We will in this paper rather pursue and propose an alternative operating point of "1/4 knot at 1/16 Watt", which substantially increases the endurance of gliders. Provided that costs are comparable, if the endurance of gliders could match that of floats then we would expect gliders to match floats in numbers and application.

## 2 Fundamental considerations

### 2.1 Energy

Consider first the profiling vehicle (float or glider) of volume $V_0$ at rest at a certain depth or pressure $p_0$. The ascent toward the ocean surface is initiated by increasing the volume by $\Delta V_0$, typically by pumping fluid from an internal to an external reservoir. This initial pumping supplies all the energy, $p_0 \Delta V_0$, needed to propel the vehicle to the surface.

However, due to ocean stratification additional pumping is necessary as the vehicle rises to maintain the initial excess buoyancy $\Delta V_0$. Most of the in-situ ocean stratification is due to the compressibility of sea water, but temperature and salinity changes also contribute. The total energy thus expended may be expressed as:

$$E = p_0 \Delta V_0 + \frac{1}{2} V_0 p_0^2 (\kappa_w - \kappa_h) + V_0 \int_0^{p_0} p \frac{1}{\rho_0} \frac{d\sigma_t}{dp} \, dp \,, \tag{1}$$

where $\kappa_w$ and $\kappa_h$ are the compressibility of sea water and vehicle hull respectively. The last term shows how ocean stratification from temperature and salinity, expressed by the density anomaly $\sigma_t$, costs additional energy. Note that this is also one of the main parameters we seek to measure during profiling.

5    Equation (1) ignores the effect of hull thermal expansion since it is small compared to sea water thermal expansion (but not negligible depending on hull material). An exact equation must include the full equation of state (EOS) of both seawater and the vehicle hull (but then becomes intractable). Furthermore, all terms and integrands must be weighted with the efficiency of the buoyancy engine of a particular vehicle.

10   We have also only stated the energy usage for the ascent part of the dive-climb cycle, which is essentially the same for both floats and gliders. Gliders generally operate in a symmetric mode in which the glider arrives at the target profile depth with a negative buoyancy equal to that used for the ascent ($\Delta V_0$). Due to compressibility effects, descending floats typically settle more gradually at the target depth (Davis et al., 1992), possibly also with a pause or parking at an intermediate depth as is done with Argo floats (Argo, 2019a).

In the following we will assume a pump efficiency as indicated in Fig. 1. This is similar to pump efficiencies reported by Davis et al. (1992) and Kobayashi et al (2010).

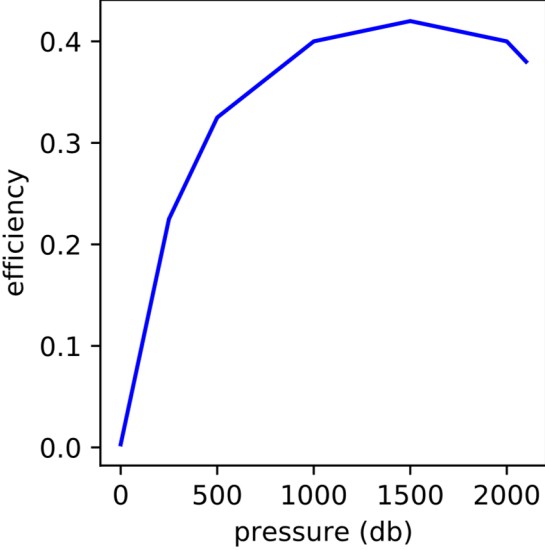

**Figure 1. Typical buoyancy engine efficiency (electric to p-V-work) variation with profiling pressure.**

Further we will assume an aluminium hull with volumetric coefficient of thermal expansion of $7\times10^{-5}$ °C$^{-1}$, and a compressibility which is 90% of that of seawater ($4.42\times10^{-6}$ db$^{-1}$). For the vehicle hull we may then use the following EOS:

$$V(p,T) = V_0 \left(1 - \kappa_h(p - p_0) + \alpha_{T,h}(T - T_0)\right), \tag{2}$$

where $\alpha_{T,h}$ is the volumetric thermal expansion coefficient of the hull and $p_0$ and $T_0$ are a reference pressure and temperature respectively. The EOS for seawater is given by TEOS-10 (IOC, 2010).

As an example, we calculate the energy consumed to ascend a tropical Atlantic profile from the World Ocean Atlas 2018 (Locarnini et al., 2018; Zweng et al., 2018) as shown in Fig 2. Vehicle volume is set at 25 L.

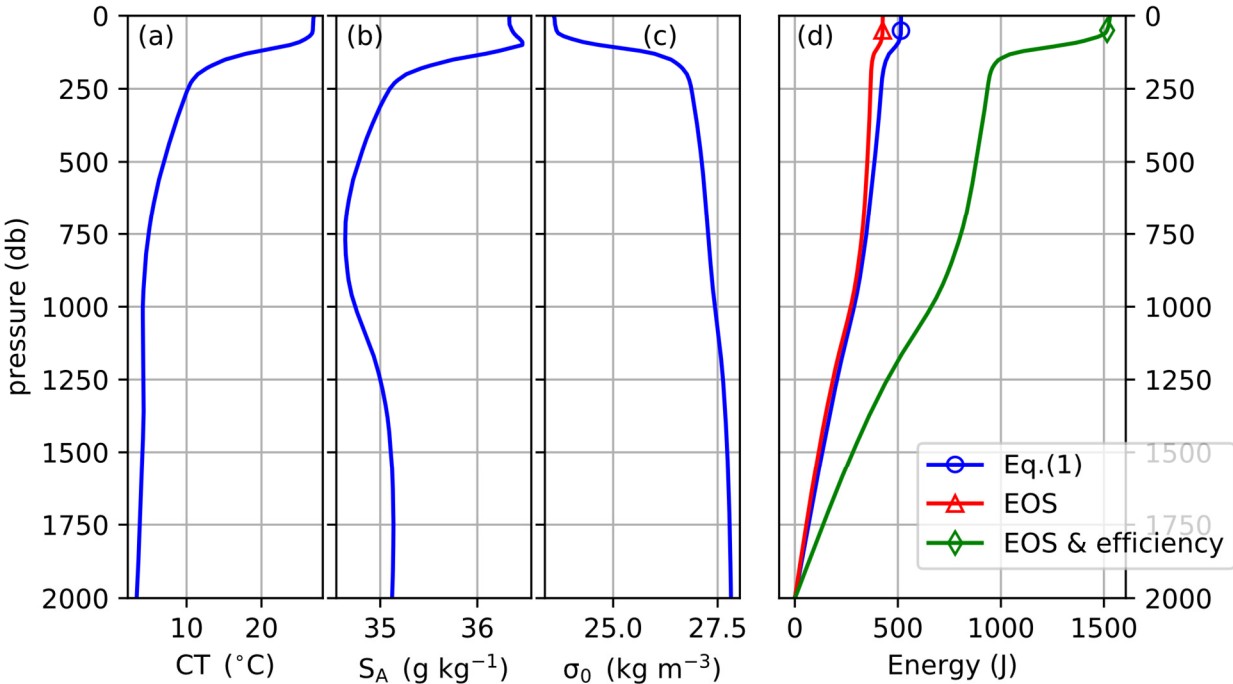

**Figure 2. Profiles of a) conservative temperature, b) absolute salinity, c) potential density anomaly and d) energy consumed. The**
10 **energy shown is that required to ascend from 2000 db toward surface estimated by the two last terms of Eq. (1) (blue), calculated from the full EOS of both water and hull (red), and also accounting for the pump efficiency from Fig. 1 (green). The example profile is from the tropical Atlantic (WOA18, 5.5°S, 25.5°W).**

Equation (1) approximates the energy consumption well, but slightly overestimates compared to the full EOS formulation. The
15 difference (compare blue and red lines in Fig 2d) is primarily due to Eq. (1) neglecting the thermal expansion of the hull (which will assist the vehicle in reaching the surface).

## 2.2 Drag and power

The drag force acting on the vehicle may be expressed as (Khoury and Gillett, 1999):

$$\boldsymbol{F}_D = \frac{1}{2}\rho V_0^{2/3} C_{DV} |\boldsymbol{v}| \boldsymbol{v} \,, \tag{3}$$

where $\rho$ is the density and $C_{DV}$ is the volume-based surface area referenced coefficient of drag and $\boldsymbol{v}$ is the 3D velocity vector of the vehicle. The area, $V_0^{2/3}$, can be replaced by any other reference area deemed suitable, such as the cross-section, length-squared, or as commonly done in aircraft design, the planform area of the wings (Hoerner, 1965). We choose the volume-based area since we expect drag to be dominated by skin friction which would scale with the wetted surface area. It should be noted that different shapes will have different $C_{DV}$'s, but for a given shape, Eq. (3) is also indicative of the scaling of drag with vehicle volume.

Power required, i.e. the product of force and velocity, is thus:

$$P = \frac{1}{2}\rho V_0^{2/3} C_{DV} |\boldsymbol{v}|^3 \,, \tag{4}$$

which was advertised in the introduction for the operation point "1/4 knot at 1/16 Watt". We will use 13 cm s$^{-1}$ (1/4 knot) as a reference velocity for the rest of the paper (note that we refer to the horizontal velocity and not $|\boldsymbol{v}|$).

## 2.3 Lift

For a winged vehicle, i.e. glider, lift is generated by the wings (Anderson, 2011; Thomas, 1999). It is known that wings are not efficient in flow with low speeds (low Reynolds numbers) (Schmitz, 1975; McMasters, 1974); however, Sunada et al. (2002) demonstrate that wings at low Reynolds numbers will perform adequately. At low speeds lift-to-drag ratios will be low (5-10), but sufficient for ocean profiling.

The generation of lift also causes so-called induced drag. In other words, the drag coefficient is also a function of the vehicle's angle relative to the direction of flow past the vehicle (the angle of attack). This effect is discussed in greater detail by Anderson (2011) and Thomas (1999), and is reasonably small here.

Categorized as flying vehicles, gliders (as discussed herein) operate in the regime of paper planes, small birds and large insects.

**2.4 Velocity and hydrodynamic model**

A hydrodynamic model is needed to calculate the vertical and horizontal components of the vehicle velocity arising from the action of the drag and lift forces. We define the hydrodynamic model in its abstract and implicit form:

5        Given expressions for vehicle drag and lift, and values for vehicle net buoyancy and orientation (pitch angle), apply Newton's first law to solve for the velocity and the angle of attack in conditions of steady planar flight.

The vehicle will then glide through the water at an angle which is the sum of pitch angle and angle of attack ($\alpha$). As each glider can have different expressions for drag and lift, and we here are concerned with a hypothetical glider, we do not elaborate

further on the hydrodynamic model and refer to Merckelbach et al., 2010, 2019; Graver, 2005 (sec. 5.1.3); Eriksen et al., 2001 and Sherman et al., 2001 for suitable expressions and parameterizations for lift and drag.

The angle of attack, however, deserves a comment in relation to lift and drag. Lift is proportional to angle of attack until the vehicle stalls and the production of lift reduces abruptly. In slow flight in particularly, caution must be exercised not to exceed

the stall angle of attack. Drag resulting from the generation of lift, i.e. induced drag, is proportional to $\alpha^2$ , hence small for small values of $\alpha$ (but not negligible).

In Fig. 3 we show the results from hydrodynamic models of two widely-used gliders, the Seaglider (Eriksen et al., 2001; Frajka-Williams et al., 2011) and the Slocum glider (Webb et al., 2001; Merckelbach et al., 2010, 2019). The Seaglider has a

relatively larger surface area hence more drag. At higher velocities and buoyancies, however, the laminar flow profile of the Seaglider improves the performance relative to the Slocum glider. We also show the performance, in terms of velocity, of a hypothetical Slocum glider with half the volume and 20% reduced drag. The reduction of volume is discussed in the next section. The 20% lower drag is justified since Eq. (3) indicates a drag reduction of 37% for a vehicle with half the volume. We deem 20% drag reduction to be a conservative estimate and will account for induced drag and parasitic drag from appendages

not represented in Eq. (3). The lift and size of the wings of this hypothetical glider is left unchanged, but it might be necessary to increase the size of the wings slightly, to compensate for the reduction in lift from a smaller hull (Merckelbach et al., 2010).

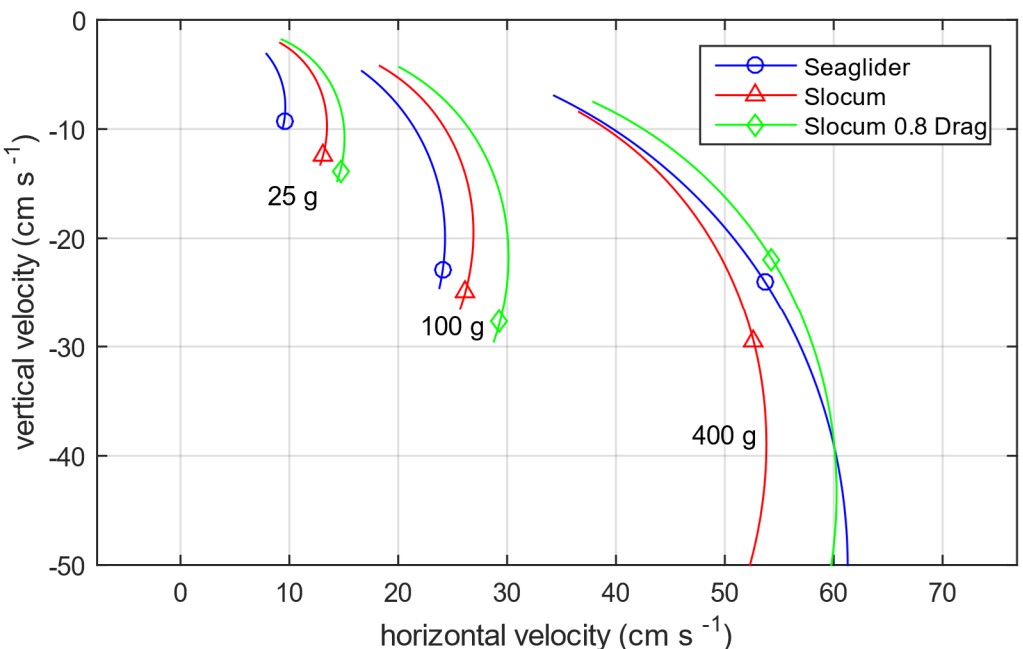

**Figure 3. Performance as reflected in velocity (speed polars) of the Seaglider and the Slocum glider for three different net buoyancies of 25 g, 100 g and 400 g. Also shown, a hypothetical Slocum glider with half volume and 20% reduced drag. At low vertical speeds, the polars are cut off at an angle of attack of 5 degrees. The angle of attack decreases with increasing vertical velocity.**

Our discussion about the performance of a glider with smaller volume is preliminary. A careful glider design should include simulations (Lidtke et al., 2018), tank tests (Sherman et al., 2001), tank-tests in combination with simulations (Jagadeesh et al., 2009), and field tests where velocities are measured (Eriksen et al., 2001; Merckelbach et al., 2019). We find the extrapolation for the hypothetical glider toward a slower velocity and lower buoyancy to be safe, and expect no significant Reynolds number effects, neither on lift nor drag. For speeds of $O(10 \text{ cm s}^{-1})$, glider Reynolds numbers are of order $10^4$ and $10^5$ for wing chords (~10 cm) and vehicle lengths (~100 cm), respectively.

We see in Fig. 3 that the modified Slocum glider with 80 % drag can achieve the desired horizontal velocity of 13 cm s$^{-1}$ for a vertical velocity of 5 cm s$^{-1}$ at a net buoyancy force of 25 g (0.245 Newton), and at an acceptable angle of attack ($\alpha < 5$). This translates to a horizontal displacement through water of 2.6 m per 1 m of profile depth, and a cycle period of almost day for a 2000 m (2031 db) deep profile. The profile depth is chosen primarily to be compatible with Argo float sampling and to escape surface currents which tend to be larger than currents at depth.

The low excess buoyancy of 25 cc will be challenging to maintain over the dive in face of ocean in-situ stratification. We have expressed the energy required to maintain this excess buoyancy as a continuous function in Eq.(1). The result of the calculation is depicted in Figure 2 (last panel) as a continuous curve. A real vertical velocity / buoyancy controller will discretise this curve as needed based on the observed depth rate which might have to be monitored frequently.

## 2.5 Preliminary discussion

All propulsive energy, pressure-volume work $p_0 \Delta V_0$, initially supplied will eventually be used to overcome drag. We need not consider residual kinetic energy when the vehicle reaches the surface since it is of order 1 J at the low speeds involved here, and the terms of Eq. (1) are typically $O(1 \text{ kJ})$.

Based on the definition of work we restate Eq. (1) with the drag expression, Eq. (3), inserted and integrated over a linear path $s$ (distance) to the surface:

$$E = \tfrac{1}{2}\rho V_0^{2/3} C_{DV} |\boldsymbol{v}|^2 s + \tfrac{1}{2}V_0 p_0^2 (\kappa_w - \kappa_h) + V_0 \int_0^{p_0} p \frac{1}{\rho_0} \frac{d\sigma_t}{dp}\, dp \ , \tag{5}$$

The following considerations follow from this equation. The speed and vehicle volume should be as small as possible. The factor $V_0^{2/3}$ in the first term might indicate that larger vehicles are better (Jenkins et al., 2003). This effect, however, is reduced by the other terms which are proportional to vehicle volume, especially for a profiling vehicle which must traverse the pycnocline (third term). Hull compressibility should match that of seawater and this effect becomes increasingly important as profiling depth or pressure is increased.

The distance $s$ to the surface is simply the depth for a vertically profiling float. Gliders, with displacement in the horizontal direction, have a longer path depending on the glide angle. Thus, gliding inherently is a costlier endeavour than profiling vertically. Also, the equation indicates that the drag of the floats should be reduced for further energy savings. A stability disc was introduced to Argo primarily to ensure better stability and communication at the surface (Davis et al., 1992); however, it turns the float into a hydrodynamically blunt object. The stability disc is not needed on floats with faster telemetry and can be removed to lower the drag and energy consumed. Glider wings suppress heaving motions at the surface, offering relatively stable communication conditions.

Equation (5) in itself indicates no optimum and instead a viable low energy consumption must be sought. Considerations including so-called hotelling loads arising from the energy consumed by sensors may introduce optima (Graver, 2005 sec. 7.2.1; Jenkins et al., 2003), but fall outside of the scope of this paper. For the vision presented here, power-hungry sensors must be avoided. It is doubtful whether a pumped C-T system could be employed on a slow glider. Unpumped conductivity cells have been successfully used in gliders, and after appropriate corrections (Lueck and Picklo, 1990; Garau et al., 2011) supply data of adequate quality. Such corrections will be challenging for a relatively slow flow past the sensor in a slow glider,

but technically possible, provided an adequate sampling rate and flushing of the conductivity cell (K. Martini, personal communication, 2019). Further calibrations and bias removal will also be possible against Argo floats and ship-based measurements. The user must carefully assess the accuracy needed for salinity against a trade off from endurance.

A net buoyancy change of 50 cc at the transition from descent to ascent at 2000 db will consume 2.5 kJ, assuming a pump efficiency of 40% (Davis et al., 1992). Assuming that the dive-climb and turning behaviour of a 1000-m rated Seaglider is representative for a slow glider, based on data from our glider missions operated from Bergen (next section), we estimate that heading control will consume approximately 1 kJ per dive for the compass electronics and the roll or yaw mechanics. Since 1/16 Watt corresponds to 5.4 kJ day$^{-1}$, there will be 1.9 kJ remaining to expend on vehicle compressibility and ocean stratification (in a dive-climb cycle period of one day). The power rating of 1/16 W is equivalent to 1.6 kg year$^{-1}$ of lithium primary batteries[1]. These numbers are for vehicle propulsion and heading control only, and an operational glider should allow for an additional 1/16 W for sensors and communications.

Present gliders indeed look compact and crammed on the inside. Yet, Eq. (5) clearly shows that volume drives energy consumption. As energy considerations are of prime importance, vehicle volume must come down. This would be achievable if the glider was designed with this consideration in mind from the start. This direction of development is necessary on the grounds of basic energy considerations. An example of a low volume vehicle is the SOLO-II float which has a volume of approximately 18 L – in its previous technological iteration, the SOLO-I float, it had a volume of 30 L (Owens et al., 2012). Reduction of volume seems possible. If glider volume could only be reduced to 30 L rather than 25 L, Eq. (5), being almost linear in volume, shows that volume and energy consumption would both increase by roughly 20 %.

The 2000 m hull of the vehicle must satisfy three requirements. It must be strong enough to withstand pressure, yet the compressibility should match that of sea water, and finally, offer the necessary payload volume for batteries, electronics and the buoyancy engine. This poses a real engineering challenge. Jenkins et al (2003, Sect. 6.3) contains detailed considerations for an aluminium hull. However, it is likely that alternative composite materials must be considered for the hull (Osse and Eriksen, 2007; Webb, 2005).

In summary, we suggest that a slow glider (or float with wings) is feasible if the volume and speed are halved relative to present gliders.

---

[1] Based on the current specification of the Electrochem 3B0036 DD Lithium primary cell:
https://electrochemsolutions.com/wp-content/uploads/sites/3/2017/05/3B0036_Datasheet-2017.pdf
Which is rated at 26 Ah, 3.2 V @ 1 A discharge, derated 10% for operation at 0 °C, cell mass of 213 g. This gives a specific energy content of 1.27 MJ kg$^{-1}$.

## 2.6 Overall power budget

As an example of a complete power budget we use a low power and slow Seaglider dive. The dive was conducted in the Iceland Sea by Seaglider sg564 on 5 November 2015 (dive number 227). The vehicle was diving with a buoyancy of ± 21 cc only, and the average vertical velocity was 5 cm s$^{-1}$. The horizontal velocity was only approximately 8.5 cm s$^{-1}$ which is 35 % slower than the velocity (13 cm s$^{-1}$) advocated by us (Figure 3).

**Table 1.** Energy/Power breakdown for low power Seaglider dive to 1000 m. Dive buoyancy was only ± 21 cc, and dive duration was 11 h. In total 860 CTD samples were collected (non-uniform vertical sampling with 20 s sampling rate in the upper 150 m).

| Main component | Parts /(subcomponent) | energy (J) | power (mW) | fraction (%) |
|---|---|---|---|---|
| Buoyancy Engine | At inflection/apogee | 1172 | 30 | 22 |
| | Stratification | 282 | 7 | 5 |
| | At surface | 179 | 5 | 3 |
| | Sum | 1633 | 41 | 30 |
| Attitude mechanics and sensor | Roll motor | 122 | 3 | 2 |
| | Pitch motor | 82 | 2 | 2 |
| | Attitude sensor | 210 | 5 | 4 |
| | Sum | 414 | 10 | 8 |
| Controller | Active (sampling, vehicle ctrl., etc.) | 1246 | 31 | 23 |
| | Sleeping | 782 | 20 | 14 |
| | Sum | 2028 | 51 | 37 |
| Sensors | Temperature and conductivity | 149 | 4 | 3 |
| | Depth (+ analog circuits) | 172 | 4 | 3 |
| | Sum | 321 | 8 | 6 |
| Telemetry | GPS and Iridium | 1014 | 26 | 19 |
| Total | | 5410 | 136 | 100 |

The controller (processor) is the most power-hungry main component with 37 % of the total energy expenditure (Table 1). This, however, is not because of complex control, but rather due to the fact that the processor of the glider is severely outdated. The controller of both Seagliders and Slocums is based on a processor design from the 1980s (the Motorola 68000-series) in a 1990s package (the Persistor). Based on a conservative application of Moore's Law, we estimate that the power consumption could be reduced by a factor of 4 for a modern processor.

Only 6 % of the total energy was expended on the CTD sensor – a figure that should arguably be increased in order to apply appropriate corrections for free-flush conductivity cells. We would like to allocate savings from a new controller to increasing the number of CTD samples possibly including an $O_2$ optode (0.7 J sample$^{-1}$).

In this paper, we are mainly concerned with the energy expended by the buoyancy engine (Eq.(1) and Eq.(5)). Nevertheless, we allow for an additional 1 kJ per 2000 m dive to be allotted to vehicle heading and attitude control. This is justified by the fact that only 414 J were expended on this during the example 1000 m dive.

Power budgets will be related to the vehicle volume as the displacement must make up for the weight of batteries. If we allocate 1/16$^{th}$ of a Watt (63 mW) to vehicle propulsion and heading control and another 1/16$^{th}$ of a Watt to the controller, sensors and telemetry, that would correspond to a 6.2 kg lithium battery pack for a two-year mission. Although challenging, it is possible to fit this battery into a vehicle with a displacement of 25 L. Please note how the example dive just falls slightly short of achieving the goal of 2/16$^{th}$ of a Watt (125 mW).

## 2.7 Mission cost

As a basis for estimating the mission cost we use the current costs for a core Argo float mission. The cost for the float itself is about 20 kUSD which approximately doubles when program management costs are included (Argo, 2019b). Basing the cost estimate on Argo float costs can be justified for two reasons. The economy of scale for $O(1000)$ slow gliders would approach that of floats rather than present gliders, and a winged float has many parts in common with regular floats; the hull, the buoyancy engine, GPS, Iridium, CTD, etc.

In Table 2 we include the additional costs for various glider specific items. A glider is inherently a more complex instrument than just a float with wings plus other components, and we also allow for costs associated with the increase in complexity of integrating the additional parts. Furthermore, we include a healthy profit of 50 % and development costs. While the relative distribution of profit, component costs and operation costs can be different, the overall cost estimate is deemed representative.

**Table 2.** Cost estimate for a slow glider mission based on Argo float costs and Argo program costs.

| Item | cost (kUSD) |
|---|---:|
| Core Argo float | 20 |
| Wings and fins | 1 |
| Roll and pitch assy | 5 |
| Attitude sensor and altimeter | 3 |
| Lager batteries | 3 |
| Complexity of integration | 10 |
| Profit of 50% on above | 21 |
| Amortization of dev. costs | 10 |
| Vehicle price | 73 |
| | |
| Argo program and data mgmt. | 20 |
| Mgmt. of complex program and data | 10 |
| Piloting (semi-automatic) | 10 |
| Launch | 5 |
| Recovery | 10 |
| Value of recovered vehicle | -10 |
| Program cost | 45 |
| | |
| Mission cost | 118 |

The simple budget in Table 2 indicates that a slow glider (winged float) mission would cost about 3 times as much as an Argo float mission (40 kUSD). This may or may not be deemed prohibitive depending on scientific potential and value of such an endeavour.

## 3. Experimental glider trajectory simulation

What missions would be possible with a glider traveling at only 13 cm s$^{-1}$? In order to explore this question, we set up a simulation experiment using slow gliders. The gliders profile to 2000 m at a vertical velocity of 5 cm s$^{-1}$ giving a cycle time of approximately one day (0.93 days to be precise).

### 3.1 Navigation

In an environment where ocean current velocities typically exceed vehicle velocity, the navigation strategy must be adjusted, or else the vehicle is simply too slow for the normal navigational notions to be feasible. Specifically, navigation with traditional latitude and longitude waypoints along straight lines must be given up. Instead, we propose to navigate in Lagrangian
streamwise coordinates.

The Lagrangian streamwise navigation is achieved when the glider steers at right angles to ocean currents, and never attempts to compensate these currents. Thus, it will be able to step into or out of any coherent current structure – be it an eddy, a front, or a boundary current. Trajectories will be spirals and oblique winding lines, and not linear transects along bathymetric
gradients. The glider will operate in a semi-Lagrangian and semi-Eulerian mode. This, however, represents a significant upgrade to Lagrangian only floats.

We call the proposed method of navigation "Eulerian roaming", where Eulerian refers to the streamline traversing capability and roaming to the Lagrangian drift. Colloquially one might be tempted to summarize the "Eulerian roaming" with two
common sayings or proverbs: "only dead fish follow the flow" and "never oppose a stronger force – out-manoeuvre". Davis et al. (2009) summarize it as follows: "*in a strong adverse current, steer rapidly across the current while making up ground where the currents are weak or favorable*". To the extent possible, in weak or favourable currents, one might still apply regular navigation. Lekien et al. (2008) address this problem.

Stommel (1989) notes: "Having to decide what heading to choose stimulated modelers and descriptive oceanographers to exercise their minds and their computers.", and we will attempt so in the following.

### 3.2 Glider simulation in a reanalysed ocean

In the simulation the gliders will attempt to navigate the reanalysed ocean of Mercator GLORYS12 provided by the Copernicus
Marine Environment Monitoring Service (CMEMS). This reanalysis is based on the real-time global ocean forecasting of CMEMS which is detailed by Lellouche et al. (2018). The reanalysis is eddy permitting with a horizontal resolution of 1/12°

(approximately 8 km) and 50 vertical levels. Temporally the output is given as daily means. For further details about the product we refer to the product user manual (CMEMS, 2018). For the purposes of the present simulation, the reanalysis need not be accurate, but should be qualitatively realistic in order to mimic the real ocean to obtain representative simulated trajectories.

A fourth-order Runge-Kutta method with adaptive timesteps (RK45) is used to integrate glider and ocean velocities to estimate the glider's position. Maximum timestep is 600 s, but this is reduced to 60 s at the surface or near the bottom, and, otherwise adjusted automatically. The glider's horizontal velocity is fixed at 13 cm s$^{-1}$ and the vertical velocity at 5 cm s$^{-1}$. The horizontal speed of 13 cm s$^{-1}$ was established in section 2.4 (Figure 3) for the operating point of 25 cc in excess buoyancy.

The velocity fields from the reanalysis product are linearly interpolated in space and time. The glider is advected in an Euclidian flat earth coordinate system, but re-projected per dive or if glider displacement exceeds 25 km. We observe no artefacts arising from the numerical scheme, linear interpolation, or spatial reference. The coarse bathymetry of the model with only 50 levels (steps increasing with depth) aggravates plunges steeper than the glider trajectory. When climbing bathymetry, the glider would occasionally fly into these plunges and get stuck, and in such cases the glider was jerked up 5 m at the time until the glider was clear of the bathymetry. This is not an issue for real gliders equipped with altimeters.

The drift at the surface, for about 5 to 10 minutes while communicating in between dives, is ignored. The results, however, are not sensitive to this.

### 3.3 Navigational recipe

The glider is steered according to the principles set out in Sect. 3.1. To express a recipe for the Eulerian roaming navigation we formulate the following pseudo code or set of rules:

A) Traverse the ocean at ±90° relative to the measured average current over the previous dive (i.e. step into or out of a certain current feature).
25
B) If depth-average current is not available, steer along or opposite to the gradient of the local bathymetry.
C) If neither A) nor B), steer to the nearest current feature as indicated by satellite altimetry (or in future, as appearing in operational ocean now- and fore-casts).
D) If none of the above provide an informed heading, use an opportunistic heading deemed suitable for the mission in
30
general.

The ordering of rules is not coincidental. They provide a hierarchy from the simplest autonomous modes (A and B[2]) to complex autonomous modes only achievable by reliance on external sensors (altimetry and models) and human or artificial intelligence

---

[2] The glider could have a bathymetric map installed to autonomously calculate the topographic gradient.

(rule C and D respectively). In the experiments presented here, we do not attempt to automate the selection of active rule, but future work must do this. For now, we rely on a skilled human pilot (a.k.a. artificial artificial intelligence).

We suppose that an up-to-date and accurate map of sea surface heights (SSH) is available, and to mimic this we use the SSH of the reanalysis as an input for the mode C above. As will be discussed in Sect. 4.5, we find this a reasonable assumption for the near future.

Occasionally the simulated glider visited ice-covered waters (eastern coast of Greenland), and we will here assume that the following under ice navigation can be executed; "head west under ice until the 500-m isobath, then turn back (without surfacing)". This can be interpreted as an under-ice version of rule B above. Gliders today are equipped with ice-avoidance algorithms (Renfrew et al., 2019), which make similar scenarios applicable.

## 4 Results and discussion

### 4.1 The Nordic Seas

To test the slow glider, we first simulate a mission in the Nordic Seas where we attempt to visit known features and currents. The Nordic Seas are bounded by Norway, Greenland, the Greenland-Scotland Ridge in the south, and Fram Strait in the north.

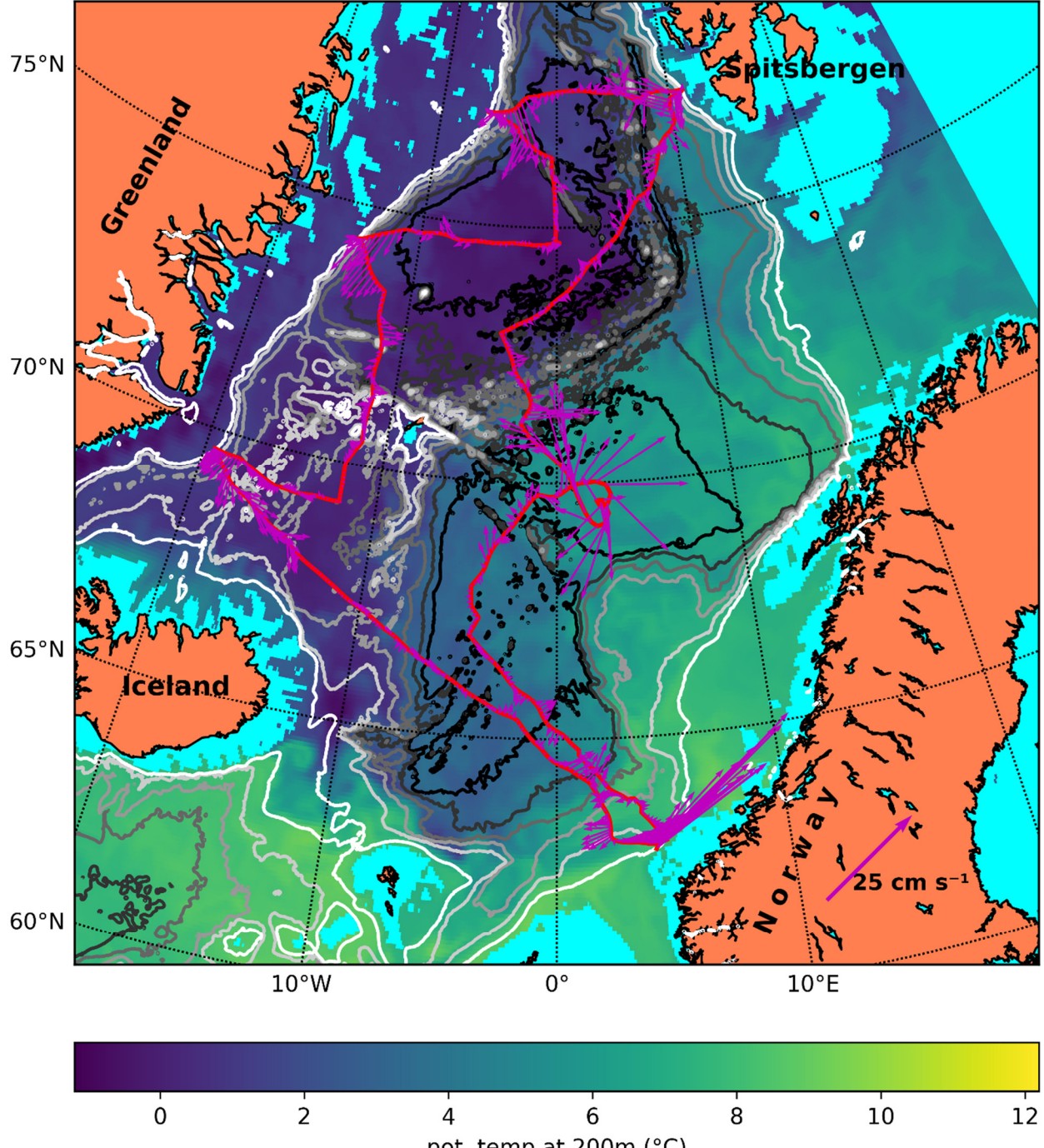

**Figure 4. Slow glider mission in the Nordic Seas.** The glider mission starts at the south-eastern corner, off Norway at 62.8°N 4.25°E at the 500-m isobath and returns 1.5 years later. The glider track proceeds counter-clockwise. Bathymetric contours are drawn at 500 m intervals. Arrows indicate depth-averaged currents measured/experienced by the glider (e.g. Rudnick et al., 2018). The temperature at 200 m is also shown to indicate water mass distribution: T > 4 °C is typically Atlantic Water with S > 35. Note that the temperature at 200 m introduces an implicit isobath at 200 m leaving shelves in a light blue colour.

The mission, Fig. 4, starts off the west-cape of Norway in the south at the 500-m isobath (62.8°N, 4.25°E, 24 June 2015 at noon UTC). The glider first heads NW off the shelf-slope. In the middle of the Norwegian Sea the glider heads NE into the Lofoten Basin where it visits the semi-permanent anticyclone, the Lofoten Basin Eddy (Yu et al., 2017). It then crosses the Mohn Ridge into the Greenland Sea, proceeds NE to Spitsbergen where it turns westward at the 500-m isobath. Unless otherwise noted, we always turn the glider near the continental shelf break, at the 500-m isobath. The glider then crosses Fram Strait westward to Greenland, and then proceeds south to the Greenland Sea. After visiting the east Greenland shelf again, the glider heads for the Iceland Sea, works another section toward the Greenland shelf, and heads SE to cross the Iceland and Norwegian Seas, reaching the recovery point where it was launched after 1.5 years at sea.

The mission executed can be summarized as follows: visit the main features of the Nordic Seas (excluding the shallow Barents Sea). Due to the relatively modest currents encountered we find that we may "ferry" the glider around according to rule D (Sect. 3.3) in the central parts of the basins. Near boundaries we used rules A and B, which often resulted in the same heading.

The glider performs 691 cycles. The energy consumption, using the technique and values described in Sect. 2, is 2.9 MJ (or 2.3 kg of Lithium primary batteries). This is calculated by evaluating Eq.(1) using the established operating point with an excess buoyancy of 25 cc and using the salinity and temperature fields of the reanalysis product. Then 1 kJ is added per dive for heading/attitude control and finally 0.5 kJ is added for surface pumping to raise the antenna out of the water. The full EOS of water and hull (Eq.(2)) is taken into consideration. Values for compressibility and thermal expansion are as given in Section 2.1 and the result of the calculation is depicted in Figure 2 panel d).

## 4.2 Gulf Stream

In order to test the slow glider in a more challenging, energetic environment, we visit the Gulf Stream.

This mission, Fig. 5, starts at the coast of Florida/Georgia (again at the 500-m isobath, 30°N, 80°W, 27 September 2015, noon UTC). The glider is rapidly advected NE by the strong currents, but is able to probe the Gulf Stream twice before it, together with the Gulf Stream, leaves the shelf break (35°N). The rapid advection with the stream continues, but not uncontrollably: at 66°W the glider intentionally visits a cold-core ring. Around 56°W the glider is caught up in an energetic meander of the Gulf Stream. The time and location of where the glider was ejected out of this meander was somewhat coincidental.

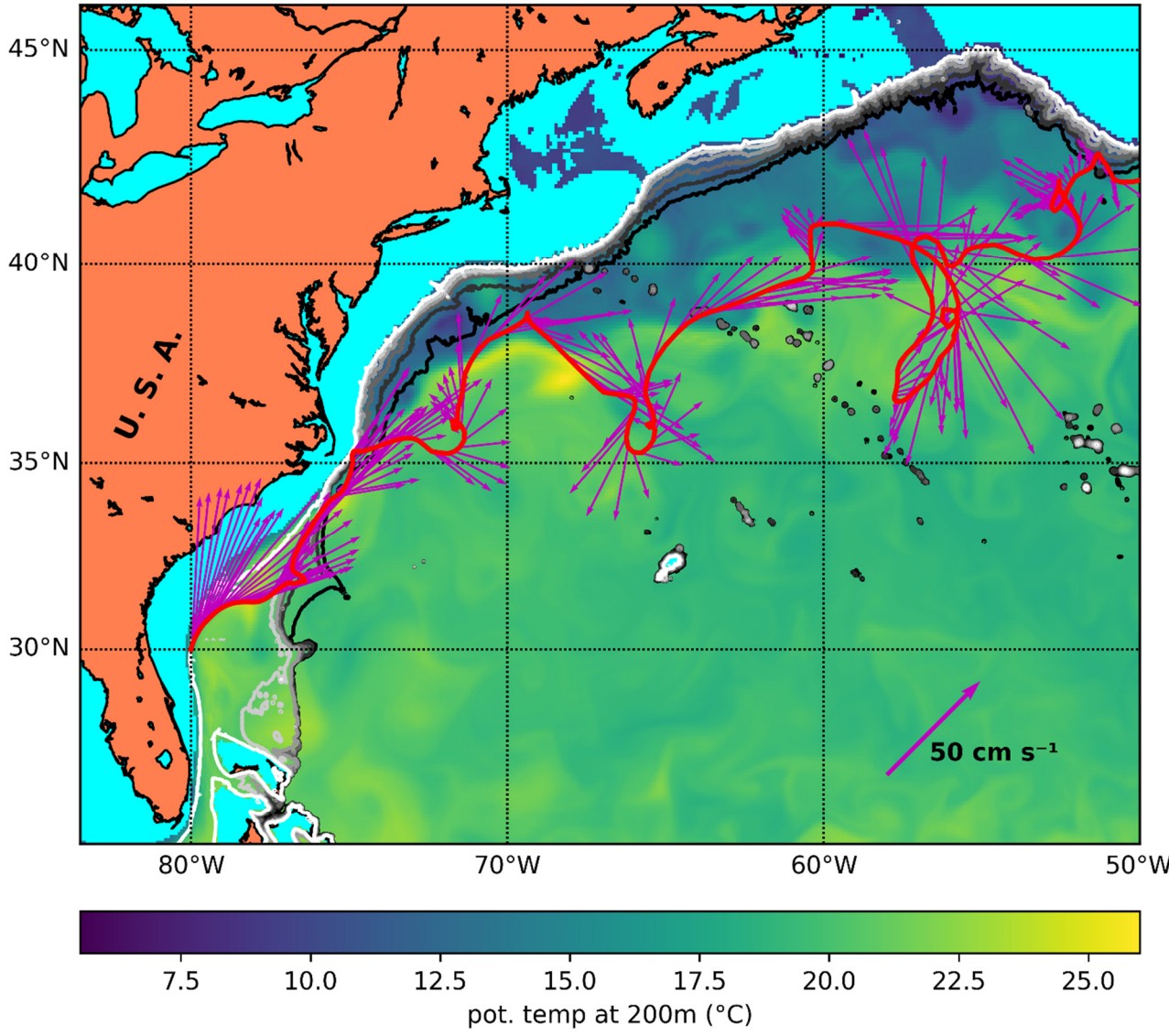

**Figure 5. Slow glider in the Gulf Stream off the east coast of the USA. The glider starts at the coast of Florida at the 500 m-isobath. Bathymetric contours drawn at 500 m intervals to 3000 m. The temperature at 200 m is also shown to represent the water mass distribution.**

The Eulerian roaming through this energetic environment is realistic and has been successfully performed previously. Using Spray gliders, Todd et al. (2016) collected transects across the Loop Current in the Gulf of Mexico, and across the Gulf Stream between 35-41°N, downstream of Cape Hatteras. To collect these sections, the gliders were instructed to attempt to fly at right

angles to the measured flow (horizontal speed of the Spray glider through the water was approximately 25 cm s$^{-1}$, the vertically averaged speed of the western boundary current regularly exceeded 1 m s$^{-1}$). The Spray gliders were operated to a maximum 1000 m depth, using the so-called "current-crossing navigation mode", in which the glider adjusts its heading after each dive to steer a fixed direction relative to measured depth-averaged currents (Todd et al., 2016). This navigation mode is similar to our rule A. It is thus demonstrated that a glider can persistently progress across a strong and variable current without continuous intervention of a pilot.

Since our hypothetical glider ended up off Newfoundland, it was natural to continue the mission into the northern branch of the North Atlantic Current (NAC), and the continuation of the mission is shown in Fig. 6.

At 55°N the decision is made to visit the southern tip of Greenland rather than continuing up along the Reykjanes Ridge to Iceland. From the southern tip of Greenland, it would be possible to work the Subpolar Gyre, but we opted to head for recovery at Iceland where the glider arrives after 721 cycles, after 1.8 years. Energy consumption is estimated at 3.4 MJ, equivalent to approximately 2.7 kg of Lithium primary batteries.

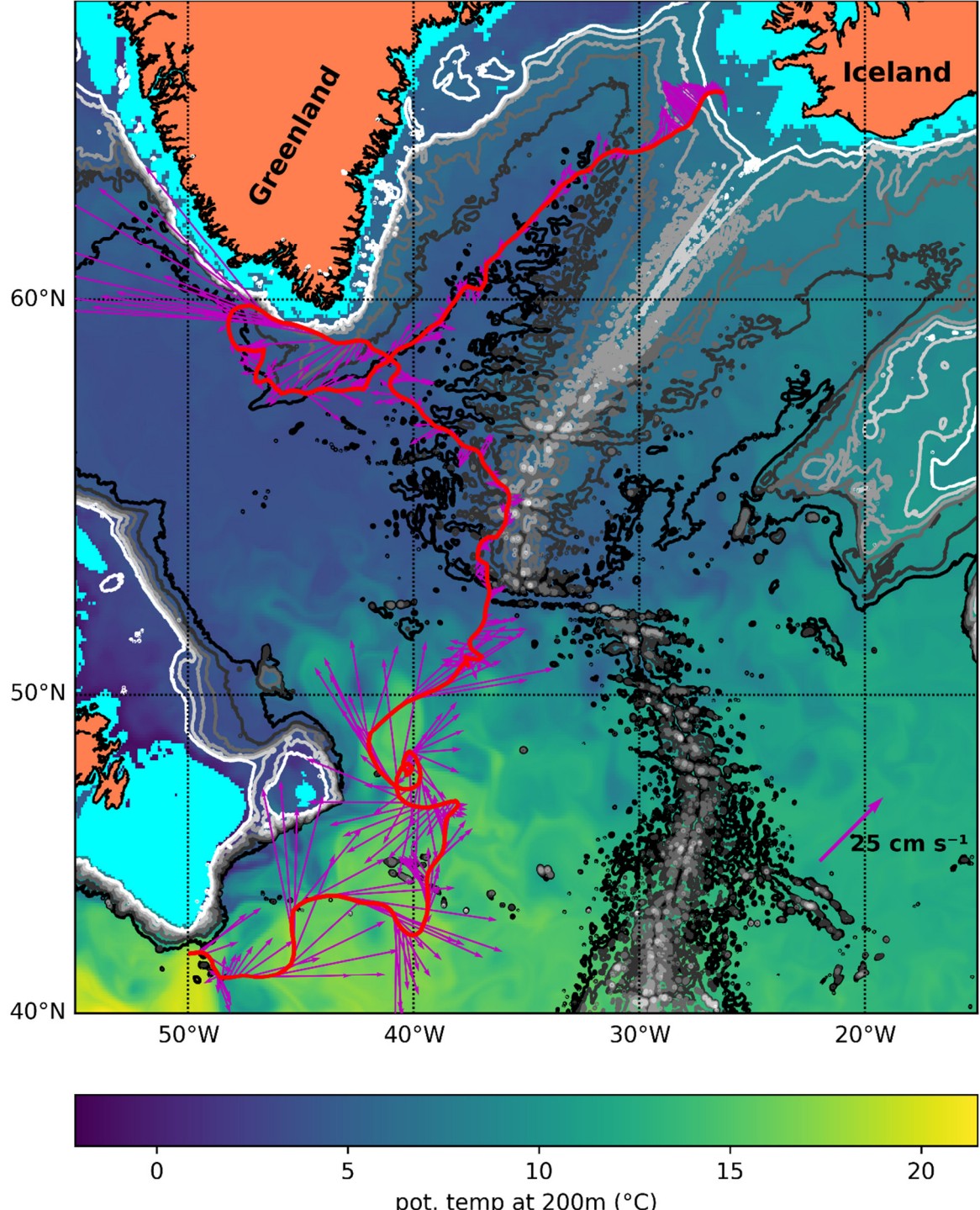

**Figure 6. Slow glider mission continued into the North Atlantic Current. The glider ends at Iceland in the north-eastern corner of the map. Bathymetric contours are at 500 m intervals. The temperature at 200 m is also shown.**

**4.3 Drake Passage**

The Drake Passage between the South-American and the Antarctic continents probably represents the world's most interesting choke point (or area) as the Antarctic Circumpolar Current (ACC) must pass through it, and we simulate a mission here as well.

The glider is launched off the tip of South America (67.8°W, 56.97°S, 26 May 2015), and attempts a transect southward across the Drake Passage. For the Drake Passage part of this mission, we attempt to do linear transects with a direct crossing of the passage. However, the slow glider is advected out of the passage in two transects (Fig. 7). This is due to the general flow of the ACC in the passage - it is simply not possible to perform a transect without drift-off here with a slow glider.

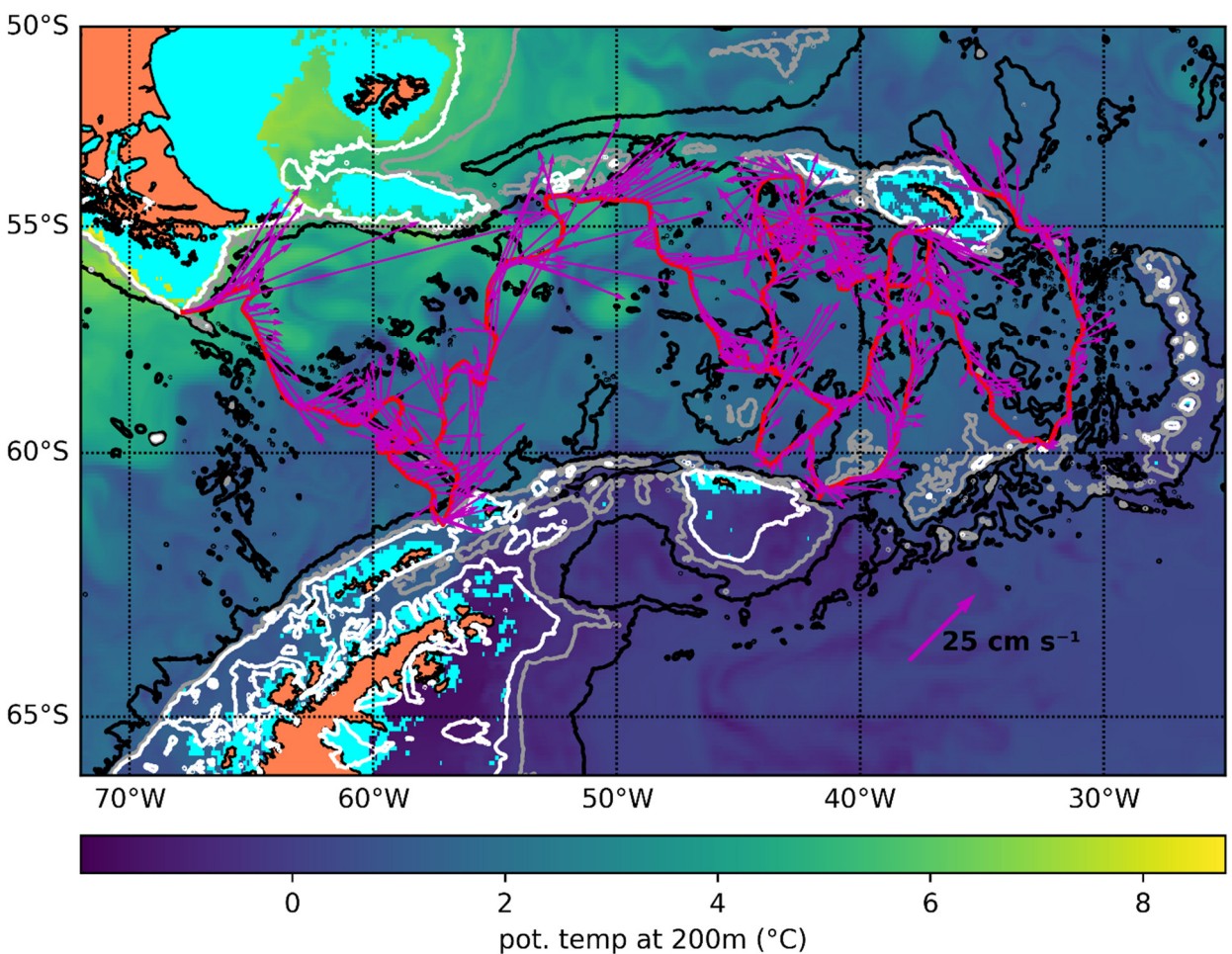

**Figure 7. Slow glider trajectory in the Drake Passage, launched off the tip of South-America (NW corner). Bathymetric contours are at 500 m, 1500 m and 3000 m. Mission ends at South Georgia Island.**

After being advected out of the Drake Passage, the glider is capable of staying in the Scotia Sea to the east, where it executes a distorted butterfly before recovery at South Georgia Island after 638 cycles. Energy consumption is estimated at 3.1 MJ (2.5 kg of Lithium primary batteries).

## 4.4 Discussion and summary

In the Nordic Seas, the slow roaming glider or winged float would significantly complement the Argo float array in the area. The slow glider is able to sample fronts, eddies and boundary currents as well as basin interiors, whereas Argo floats tend to be constrained within the 2000-m isobath of the basin where they were launched (Voet et al., 2010).

The mission exemplified in the Nordic Seas targets the observation of the circulation and water mass properties at key locations in the Nordic Seas. This region is a key component of the Atlantic Meridional Overturning Circulation (AMOC) in which warm waters flow northward near the surface and cold waters return equatorward at depth. The variability of the Atlantic Water characteristics is of importance to the climate in western Europe, weather and sea ice conditions, on primary production and fish habitats. The Nordic Seas are an important area for water mass transformation (Mauritsen et al., 1996, Isachsen et al., 2007). The newly produced or transformed dense waters return southward between Iceland and Greenland through Denmark Strait, and east of Iceland across the Greenland-Scotland Ridge, contributing to the lower limb of the AMOC. Transects worked by a slow glider will provide crucial observations in the Norwegian Atlantic Current at Svinøy (Høydalsvik et al., 2013), in the deep convection regions in the Greenland and Iceland Seas (Brakstad et al., 2019; Våge et al., 2018), and in the Lofoten Basin which is a hotspot for Atlantic water transformation (Bosse et al., 2018). The transect in Fram Strait will capture the properties and variability in the return Atlantic Water along the Polar Front in the northern Nordic Seas (de Steur et al., 2014). Particularly the interior Greenland and Iceland Seas, and the east Greenland shelf are under-sampled, and the observations will be useful in understanding the role of wintertime open ocean convection in the western basins of the Nordic Seas and the effect of an ice edge in retreat toward Greenland (Moore et al., 2015; Våge et al., 2018).

Similarly, slow glider observations from the Gulf Stream and the Drake Passage mission examples will advance characterization of mean pathways, mesoscale variability and energetics in climatologically important regions. Furthermore, the Eulerian roaming will allow sampling of snapshots of mesoscale eddies. In the Lofoten Basin, a similar navigation option was used to spiral in and out of the Lofoten Basin Eddy by instructing the glider to fly at a set angle from the measured depth-averaged current (Yu et al., 2017). Profiles collected from such missions will be useful in characterising the coherent eddy structures, filaments along fronts and around mesoscale eddies (see Testor et al., 2019, and the references therein). An

additional strength of glider observations is the ability to infer absolute geostrophic currents (e.g., Høydalsvik et al., 2013). The transects resulting from Eulerian roaming are different than and less regular compared to the sections occupied by ship-based surveys, typically normal to the isobath orientation. Strong current speed exceeding the speed of gliders will result in oblique sampling. However, the local streamwise coordinate system (Todd et al., 2016), applied to for instance the Gulf Stream and the Loop Current, is demonstrated to be a powerful approach to calculate volume transport rates, potential vorticity structures and provide insight into processes governing flow instabilities.

The key assumption in using the local streamwise coordinate system for geostrophic current calculations along the glider trajectory is that flow is parallel to the depth-averaged current (DAC). When the depth-average current direction is not perpendicular to the transect segment of the glider path, a decomposition into cross-track and along-track components must be made. In these conditions, using the currents from the local streamwise coordinate system will be in error; however, the transport will remain relatively unaffected. In a recent study, Bosse and Fer (2019) reported geostrophic velocities associated with the Norwegian Atlantic Front Current along the Mohn Ridge, using Seaglider data, following Todd et al. (2016) and assuming DAC is aligned with the baroclinic surface jet. They also calculated the geostrophic velocities and transports using the traditional method, i.e. across a glider track line, and found that the peak velocities of the frontal jet were 10-20 % smaller but the volume transports were identical to within error estimates. The Eulerian roaming can thus be used to obtain representative volume transport estimates of relatively well-defined currents. We also note that the present 1000-m depth capability of gliders limits our ability to compose the geostrophic currents into barotropic and baroclinic components in water depths substantially deeper than 1000 m. A 2000-m range will allow us to more reliably approximate barotropic currents as the depth averaged profile, up to depths of around 2000 m.

Regarding the navigation in general we note that it was easier and required less skill and intelligence in strong currents, when only a choice between left or right could be made (rule A), this often coincided with bathymetry (rule B). In weak currents, the navigational options increased, and we resorted to skilful and intelligent use of rules C and D. The intelligence, however, does not seem to be very advanced as it essentially is an image processing task on the SSH image (albeit a vector image) for local steering decisions. Global steering decisions such as general area to visit will require some oceanographic intelligence and is probably not suitable for automation.

The control and steering of a network or fleet of slow gliders should aim to optimize for some scientific objective possibly in conjunction with other sensing platforms. Alvarez and co-workers (2007) have looked at synergies between floats and gliders to improve reconstruction of the temperature field. Synergies also exists between a glider fleet and altimetry to map geostrophic currents (Alvarez et al., 2013). We suggest that future work should see the slow glider concept as part of a heterogenous suite of ocean sensing technologies. The topology of the network needs some consideration and one interesting option is to cluster the gliders in and near an oceanographic feature to explore it in greater detail. Some in-situ experiments with glider fleets have

been conducted (e.g. Leonard et al., 2010; Lermusiaux et al., 2017a). The problem of planning optimal paths for gliders is reviewed by Lermusiaux et al. (2017b).

While we have shown tracks of individual gliders, it should be clear that the impact of a slow roaming glider concept will increase when employed in large numbers. Also, the simulations here where a few gliders are hand-piloted does not show the full potential of the approach. Future simulations can include a large number of gliders to train artificial intelligence to perform the piloting.

## 4.5 Outlook – altimetry, models and Argo

The upcoming Surface Water and Ocean Topography (SWOT) altimetry mission (Fu and Ubelmann, 2014) will yield an unprecedented view of the ocean surface. Within the swath of the altimeter, approximately 120 km wide, we will see snapshots of oceanic mesoscale and sub-mesoscale structure and variability, albeit at a slower repeat cycle of 21 days. However, advances in processing (Ubelmann et al., 2015) will likely fill the temporal gaps in a dynamically meaningful way, leading to maps of SSH with high temporal resolution, and enable operational model capabilities and applications hereto unimagined (Bonaduce et al., 2018).

There was always a strong coupling among altimetry, models and observations of ocean interior (Le Traon, 2013). The Argo programme's name was chosen because of its affinity with the then upcoming altimetric mission of the Jason satellites. Argo was the ship of Jason and the argonauts in their quest for knowledge. As altimetry advances, it is necessary to ask whether Argo and our quest for knowledge should advance in parallel. While SWOT altimetry will yield a (sub-)mesoscale view of the ocean, Argo remains primarily a basin and seasonal scale technology. We propose that the slow glider concept, essentially the gliding float that the Argo design specification calls for, could add a mesoscale component to Argo. This natural development enhances the Argo as a component of the global ocean observation system, and supplements the regular glider operations, which are at present regional and process-oriented (Testor et al., 2009; Liblik et al., 2016; Testor et al., 2019).

Since we propose to steer the glider using maps of SSH and model output, the proposed slow glider would also provide an even tighter integration of altimetry, models and observations of ocean interior.

Other developments in the Argo programme further suggest a progression in this direction. Floats are increasingly being equipped with more advanced sensor suites in the biogeochemical program (Riser et al., 2016; Roemmich et al., 2019). A new ArgoMix component with turbulence sensors (thermistors and airfoil shear probes), is also under consideration to map the spatial and temporal patterns of ocean mixing. The capabilities of the sensors call for a more advanced vehicle navigating the mesoscale ocean as well.

In the example missions presented here the glider is launched and recovered near the coast. Logistical challenges aside, this opens up new participatory dimensions with coastal communities. It might also be judged as a more environment friendly alternative to the Argo floats which are submitted to the ocean upon mission completion.

While the tracks of the slow glider (/winged float) presented in this section clearly demonstrate oceanographic *potential* it remains to prove scientific *value* added to the existing network of Argo floats, regular gliders and altimetry. The scientific value could be explored and possibly quantified by an Observing System Simulation Experiment which would include all observing elements of the GOOS including our slow virtual glider. Such future work might build on the concepts and methods 10 presented here. In Section 2.7 we roughly estimate that slow glider missions will cost 3 times more than a float mission, which requires that the scientific value be correspondingly enhanced if Stommel's vision is to be implemented in the form of slow gliders as we propose.

## 5 Conclusions

We show that oceanographic useful and sensible trajectories are possible with a slow roaming glider. Looking back at the 15 quote from the Argo design specification in the introduction, one might say that the expectations for a gliding float were too high. The notion of "a fixed location or along a programmed track" is not feasible due to energy constraints limiting velocity, nor is the notion indispensable or necessary. Even though we here mostly explore the concept of "Eulerian roaming" navigation, the slower and smaller glider will be able to maintain station (virtual mooring) or follow well-defined section lines at sites where currents are weak.

The velocity of $25 \text{ cm s}^{-1}$ is unrealistic for endurance missions of years given the current status of battery (and/or energy harvesting) technology. The speed mentioned by Stommel (1989) in his vision, was merely an example and should not be a constraint. We have shown that $13 \text{ cm s}^{-1}$ can be sufficient to navigate the ocean giving due consideration to energy/power constraints.

Future work should firstly attempt to verify the concepts and findings presented here using existing gliders in the real ocean. The gliders should be operated at a lower speed than usual (refer to Fig. 3) and navigated as outlined in Sect. 3.1 and 3.3. Future work should also include observing system simulation experiments (e.g. L'Hévéder et al., 2013; Chapman and Sallée, 2017) whereby data assimilation from fleets of slow gliders demonstrate benefit and increased model skill in operational 30 models. The piloting should also be automated, and work might be directed at developing artificial intelligence doing day-to-day piloting.

This paper demonstrates that slow gliders or Argo floats with wings are desirable and potentially feasible – the slow glide is on.

*Data availability.* The data used in this paper are made freely available by CMEMS (CMEMS, 2018) and in the World Ocean Atlas 2018 (https://www.nodc.noaa.gov/OC5/woa18/).

*Author contribution.* E.M. Bruvik and I. Fer wrote the paper with revisions and suggestions from K. Våge and P.M. Haugan. E.M. Bruvik performed the simulation experiments.

*Competing interests.* I. Fer is a member of the editorial board of Ocean Science, but other than that the authors declare that they have no conflict of interest.

*Acknowledgements.* The glider team at the Geophysical Institute spurred many interesting discussions about gliders and glider
technology. The glider activities here were initially financed by the Research Council of Norway under grant no. 197316. Additional support for this work was provided by the Trond Mohn Foundation under Grant BFS2016REK01 (K.V.). We thank L. Merckelbach and NN, for their constructive review and criticism of our paper. Interested readers should consult their detailed remarks and consider them supplemental in the spirit of *Ocean Science Discussion.*

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
