# Peer review of "A revised ocean glider concept to realize Stommel's vision and supplement Argo floats"

_Ocean Science, 2019_

## Referee Comment (RC1) · Anonymous Referee #1 · 17 May 2019

Dear Editor,

This is a bit of an unusual manuscript. It is not very scientific in the traditional sense, but interesting nonetheless. It makes it a bit difficult to rate the questions 1) and 2) above appropriately. I attach a pdf with detailed comments and issues. Interesting as it looks, I am not really convinced if it would work out. I realize that it may be not straight forward to alleviate my concerns. Inspite of this I would not wish to mark it is as "reconsider after major revisions" and I hope it will start a discussion on this matter.

Please also note the supplement to this comment:
https://www.ocean-sci-discuss.net/os-2019-36/os-2019-36-RC1-supplement.pdf

[Figure]

**Supplement:**

**Review of**

"A revised ocean glider concept to realize Stommel's vision and supplement Argo floats" by Erik M.Bruvik, Ilker Fer, Kjetil Vage, and Peter M. Haugan.

This manuscript goes back some 30 years, when Stommel presented his vision on how the world's oceans could be sampled using a vast network of autonomous robots. The authors describe the extent to which this vision has materialised and what aspects have not been realised yet. The original concept consisted of gliding robotic underwater vehicles. The Argo program that uses vertically profiling floats. The underwater gliders that have been developed since then, never managed to get deployed in big numbers and on a global scale. The main hurdle is the limited endurance of gliders.

The authors look into the energy consumption of the underwater glider and make a case for a downscaled version of a glider, both in size and speed, that would give the glider the endurance required of the order of years, and add an active steering component to the device, which the floats, which operate in a Lagrangian fashion, lack. They simulate a hypothetical downscaled glider in some key parts of the worlds ocean, demonstrating the added value of a gliding float, which they call an Eulerian roaming float, compared to a Lagrangian float.

I enjoyed reading this manuscript. It is written in a well-structured way, and clearly presented. I don't feel qualified to suggest alternatives for grammar and spelling issues, so I refrained from listing any. In any case, I think overall the manuscript is well-written.

After reading it, I was left a bit with a "So what?" feeling. I think this might be due to two things. The authors primarily look at the energy consumption required for propulsion, to argue that a horizontal speed of 13 cm s-1 would be sufficiently low to increase the endurance level of the (smaller) glider to the time scale of a year. The energy consumption by the sensors and controllers is marked as "beyond the scope of the research". I think that this is in fact a very important aspect. And it may even be so, that the sensor/controller energy requirements are the limiting factor. The reason why I think this, is as follows. I must note that I am no expert on Argo floats, but it seems the expected life time of floats is some four years, where the primary limiting factor is depletion of the batteries. The standard cycle takes 10 days. A gliding float, would do a profile a day. Assuming that the float's controller would go into a deep sleep during the drift-at-depth phase, then profiling every day would slash the endurance down to half a year. Of course it depends on how much batteries the device contains, but this makes me wonder. You do state that based on experience with a Seaglider, a cycle takes about 1 kJ for the electronics, which then would translate to 1/16 W, and could make the concept feasible. My experience is mainly with Slocum gliders, and, although the manufacturer has done a lot to reduce the power consumption of the electronics, a figure of 1-2 W is more appropriate. So clearly, sacrifices need to be made in terms what, how and how much is measured. Contrary to the Slocum glider, whose processor is effectively constantly running, a very low power system would sleep most of the time. The fact that going so slow requires only a small buoyancy drive, it also means that stratification may cause significant changes in the effective buoyancy drive, and as a consequence may require frequent monitoring of diving or climbing rates. A similar argument goes for keeping course. If the Seaglider is indeed so efficient (again, I am out of my comfort zone here), then simply reducing the buoyancy drive would allow for existing gliders to be used as described in the manuscript. Because this whole aspect is glossed over by saying beyond the scope of the manuscript, it does not really convince me.

Another aspect that does not convince me, because it is not properly addressed, (and I agree immediately that it is not so clear how this would be addressed properly) is, if it is at all possible to

build a vehicle half the size that contains all the hardware needed to function **and** sufficient amount of batteries. If batteries is the limiting factor, a bigger glider may be more advantageous.

The final issue I have, is the costs of such a down sized glider. I reckon that a guide price of today's conventional glider glider is about \$ 200k. To be deployed in thousands, the price must come down enormously. I am skeptical about the financial viability of the design, but I would loved to be proven wrong... Weirder things have happened though.

Sections 4.1-4.3 show the results of such a glider that is deployed in various parts of the world's oceans. Personally I felt that each case conveys more or less the same message, and the one case would be as good as any other. Like one float, one glider would not tell much about the state of the ocean, and the appeal is a large number of devices. I thought that focussing on one case, where you look at how the added information of a Eulerian-roaming device, as opposed to a Lagrangian device would give, would be more compelling.

I realise I sound a bit negative, and I also admit I also don't really know how you could factor in the issues I raised.

**Detailed remarks**

Below I list a number of remarks I made in the margin when reading the manuscript.

P1. L23: "in that sense fall short of realizing his vision", this sentence suggests that as long as it has wings, all is well. I think what you mean is that the dynamic positioning is what is failing.

P2. L4 "simpler", simpler than what? Also related to this paragraph is that "just adding wings to a float" in reality comes with a serious increase in the level of complexity.

P3. L16. This sentence initially confused me, but it made sense after I looked up some details of the Argo float. I think the words "pause" and "parking" in this context are not clear for someone who is not very familiar with how floats are typically operated.

p.6 L9 A considerable part of the lift is generated by the hull of the glider.

p.7. L14-16. (Related to the previous point) "... to compensate for .... smaller hull": this suggest that, at least for the Slocum gliders, the design of the wings (size/shape) is somehow optimized. I suspect it is not, as the leading principle in the design of the Slocum glider is easy construction and I don't think much thought has gone into the size of the wings.

p.8 L15. Here I thought is might be difficult to achieve gliding with just 25 g of buoyancy, because the effective buoyancy may change more than this in a stratified ocean, and using a vehicle that has not a compressibility that is exactly matching that of seawater. This requires frequently adjusting the buoyancy on both the down and up casts.

In my experience gliders typically reduce speed on both the down and up casts, due to stratification effects. You could, I suppose, store energy when reducing the volume on the own cast and releasing it again on the upcast, but still you will look at a hysteresis-like effect, and a much more technically complicated design (read: losing volume for batteries, increased costs).

p.11 L19 Here you say you set the glider's velocity vector. It is not clear to me where you specified the just that, the speed, or that you would specify the buoyancy of 25 cc. I guess you prescribed the speed. In that case, my previous point should some how be addressed. If you specified the buoyancy, I suggest you include a small discussion on how frequently the buoyancy needs to be changed, and what the energetic costs are.

p.14 L12. Here (and also in following paragraphs) you specify the energy consumption. I read this number as 691 cycles, at 1 cycle a day, using 1/16 W gives 2.9 MJ. Or is this computed using equation 5, taking into account the actual velocity the glider made, and stratification it faced, and the effort done to compensate for it? Also I think, this includes only the power required for propulsion. So what about the electronics?

---

## Author Comment (AC1) · 24 May 2019

Dear Editor and Referee #1,

below we offer a quick reply in order to facilitate discussion rather than a full answer to the review.

Reviewer #1(R#1) correctly points out that this is a somewhat unusual manuscript. This partially stems from the fact that we build on Stommel's vision for a global glider network. Stommel (1989) presented this vision as a short science fiction story. In our reappraisal we do take a less experimental approach genre-wise, even standard scientific we would argue, by first putting forward the equations for the energy consumption needed to propel/move the glider through the water. This equation (5) shows that a

smaller and slower vehicle would consume less energy (to move through water). We then conduct some (simulation) experiments to show that a slow glider could indeed navigate the ocean in a meaningful way.

R#1 gives a good synopsis of the paper and we are thankful for his/her kind efforts with this unusual, but still comprehensible paper. From the point of understanding and appreciation of the text R#1 does raise relevant and substantial concerns. We find these to have two foci: A) Would/could it really work? Questions and doubts of technical feasibility. B) We do not properly address the total energy consumption of a complete glider with controllers and sensors.

Ideally, we would have a complete, actual vehicle to show for, but instead we are forced to make considerations about a future hypothetical glider. R#1 will certainly appreciate that the design of such a vehicle would require a substantial human and economic effort beyond our resources.

As far as concern A) is considered, let us put forward the following argument: both floats and gliders are already feasible technological facts. A hybrid of the two technologies, a slow glider or float-with-wings, seems doable as it might be thought of as a "mean" of the two classes of vehicles (this is of course a simplification). We are in the paper thus looking at a technological interpolation and not an a very speculative extrapolation.

R#1, for instance, expresses doubts that a small glider with a volume of 25L is technically possible. And we agree that present gliders indeed look compact and crammed enough already on the inside. Yet, Eq. (5) clearly shows that volume drives energy consumption. As energy considerations are of prime importance, we believe that vehicle volume must come down. And that this is not impossible if the glider was designed with this consideration in mind. This direction of development is necessary on the grounds of basic energy considerations. An example of a low volume vehicle is the SOLO-II float which has a volume of approximately 18L – in its previous technological iteration, the SOLO-I float, it had a volume of 30L (see table on second slide of http://www.argo.ucsd.edu/AST13_SOLO-II_Status.pdf). Reduction of volume thus seem doable.

Further, R#1 raises concerns about the control of the vertical speed of our conceptual glider – especially at such low buoyancy as we prescribe (25 cc). In our work this is evaluated as continuous integrals and equations (Eq.(1) and Eq.(5)), clearly, a real glider must approximate this as discrete pumps(/bleeds). We believe this control problem to be solved in the implementation of floats as they also aim for a nearly constant vertical velocity and also operate at low buoyancies. As R#1 duly notes this and similar controls will have to be finer for a slow glider and is a step up in complexity of control compared to floats.

This brings us to issue B) – the control exercised and controller itself will require some energy. It will be very difficult for us to address this issue adequately as it is highly dependent on controller implementation and dynamics of the vehicle. Similarly, sensors may use more or less energy depending on configuration, implementation and sampling. We can thus not close the power budget as both R#1 and we would like.

It should be noted that we are not oblivious to the problem. In the paper we state that: "For the vision presented here, a power-hungry sensor must be avoided. This casts doubts whether a pumped C-T system could be employed on a slow glider."

Also, we do factor in energy consumed for heading control at a ball-park value of 1 kJ per dive/cycle. We do this since it is clear that the heading/attitude sensor will require energy and also energy for the control mechanism. This number is based on our experience with running endurance missions with the Seaglider. We may substantiate this with a power analysis/breakdown of an example low power Seaglider dive if R#1 wishes (?).

One should also consider and discuss some options for the glider technology in future. One option is to proceed with status quo: floats do Argo and glider missions continue

to be more sporadic using 20 years old designs. Another option would be larger vehi-cles with more battery and more endurance. This could lead to designs which could require specialized equipment for launch and recovery – not a very feasible prospect. A third option would be to passively wait for leaps in battery technology, say, when batteries have improved 4-fold. We believe the novel concept proposed by us to be yet another prospective and more promising option than the aforementioned. The scien-tific community should know about this option and further discuss and elaborate on the concept.

---

## Referee Comment (RC2) · Anonymous Referee #2 · 8 Oct 2019

**General comments**

This is an interesting paper, looking into the concept of a smaller, slower glider from a practical sampling and operational perspective. It is clearly shown that the idea has potential from this perspective, but it has not been adequately shown that scientifically or economically, it would be a better choice than the current one. Attempts to justify the Eulerian roaming were not convincing, especially for single missions. Using a much larger number and different analysis techniques would help, but it was not justified exactly how this would work. What scientific questions beyond the lucky detection of an episode or feature would such a large scale network address that ARGO floats do not already address? What is so special about 1000 gliders exactly? Are the reasons Stommel used to justify that number still relevant today? Is a global coverage of gliders

really the most effective use of resources, or should they be used for specific purposes regionally, but in a scaleable, interoperable way? None of this is addressed. Central to this question is cost, which was essentially brushed aside.. Presently, gliders cost about 10 times more than floats to purchase. Operating costs are much higher because pilots are needed, even when the available autopiloting and fleet-type behavior is used. This will get better eventually, but at present the gross value of hardware in the water is not that different: 4000 floats vs 40 gliders is a factor of 10 of higher investment in float hardware cost compared to glider hardware, a gap which is nearly closed when considering piloting costs, increased data transmission costs, boat recovery costs, and the greater land infrastructure requirements. None of these are reduced with a smaller glider, only in the sense of economy of scale as is currently the case as well. The point is that investment in gliders globally is not far behind that of floats, but it is not coordinated globally yet. To have a fleet of 1000 would require serious increases in investment, and thus very strong scientific justification.

Specific comments

While the general comments above essentially critique the introduction and premise of the paper, some specific comments are laid out for the scientific content that follows. Section 2 Fundamental considerations: this section is scientifically sound and well written. On page 7, line 10: it may be stated that those glider manufacturers now have different designs and that performance may differ (e.g. Seaglider ogive fairing or larger Slocum G3 hull). It would be interesting to update the results for those and to run more simulations for reduced volume versions, rather than just one. page 9, line20-25: It is not clear if a CTD-only glider will best serve the global observing system: there are many more Essential Ocean Variables that gliders can (and soon will) be able to measure. This flexibility is one of the strengths of current gliders. Some examination of what payloads would be possible compared to what is normally done now would be interesting, and I think not outside the scope of the paper. Later in the paper, microstructure is mentioned. That paragraph could be expanded to include

other potential payloads for the small glider. I am not sure why detailed power budgets and engineering calculations should be excluded from the paper. It seems to me that would strongly support the main point of the paper. More details about the strengths and weaknesses of Eulerian roaming are necessary if the reader is to believe this is a viable alternative. The simulations following help, but no indications are given on how such data could be/have been handled other than a simple citation (Todd et al., 2016). This section 2.6 seems out of place, and fits better in the next section.

Section 3. Clear, but should be merged with 2.6.

Section 4. Results and Discussion. The hypothetical case studies are interesting and show the potential, but are not convincing in terms of scientific value. An attempt is made in 4.4, but the analysis from the mission is oversimplified in my opinion. Separating temporal and spatial variability on these year long missions over large horizontal gradients would be very difficult and it is not always possible with one long track to collect data "useful in understanding the role" or that will "capture the properties and variability". The section about altimetry begins to touch on what could be the scientific goal of such a fleet: the surface topography problem. The number of gliders needed to reduce the current errors in the altimetric eddy field (number, phase and intensity) could be quantified in this paper and justify the existence of the fleet.

Section 5. Specific methods of piloting large numbers should be cited (optimal fleet mission planning) as well as the scientific objectives one might achieve with this (e.g. optimized for data assimilation for altimetry or some other objective). This was very briefly touched upon in the conclusions and future work, but really this should provide a solid background to why the reader should even dig into the paper. Clearly this concept is most valuable in a complex large fleet sampling context and some work has been done already.

---

## Author Comment (AC2) · 4 Nov 2019

Dear Editor and Referees,

Thank you for handling our manuscript and providing constructive comments. Both referees find our paper interesting, well-written and well-structured. We clearly introduce a novel concept based on Stommel's vision and demonstrate its potential. However, both reviewers raise substantial issues which may be grouped as:

a) lack of complete power budgets
b) lack of evaluation of the cost aspect
c) lack of demonstration of added scientific value

To address point (c), an Observing Systems Simulation Experiment (OSSE) must be conducted, simulating 100s of gliders. We suggest that this is beyond the scope of the paper and would need to be addressed in a separate paper. Our present work is an important prerequisite, introduction of concept and motivation for such further work.

Regarding points (a) and (b), we include two new sections to the paper as described below. This is followed by our point-by-point response to the referees' comments. We outline the necessary revisions to address these comments. Given the critical reviews, we submit this final response and await the Editor's recommendation for submission of a revised manuscript to be considered for *Ocean Science*. The authors think the manuscript will be of interest to *Ocean Science* readers, and that there is sufficient insight and novelty to qualify as a scientific paper.

In the event that the manuscript remains in the archives of *Ocean Science Discussions*, the study and the concept will be accessible, citable and informative to the interested readers. Hence, we would like to offer the best possible paper and provide a revised manuscript also in this case.

Best regards,

Erik Magnus Bruvik, corresponding author

**Suggested amendments to the revised version:**

**2.7 Overall power budget**

As an example of a complete power budget we use a low power and slow Seaglider dive. The dive was conducted in the Iceland Sea by Seaglider sg564 on 5 November 2015 (dive number 227). The vehicle was diving with a buoyancy of ± 21 cc only, and the average vertical velocity was 5 cm s$^{-1}$. The horizontal velocity was only approximately 8.5 cm s$^{-1}$ which is 35 % slower than the velocity (13 cm s$^{-1}$) advocated by us (Figure 3).

**Table 1.** Energy/Power breakdown for low power Seaglider dive to 1000 m. Dive buoyancy was only ± 21 cc, and dive duration was 11 h. In total 860 CTD samples were collected.

| Main component | Parts /(subcomponent) | energy (J) | power (mW) | fraction (%) |
|---|---|---|---|---|
| Buoyancy Engine | At inflection/apogee | 1172 | 30 | 22 |
| | Stratification | 282 | 7 | 5 |
| | At surface | 179 | 5 | 3 |
| | Sum | 1633 | 41 | 30 |
| Attitude mechanics and sensor | Roll motor | 122 | 3 | 2 |
| | Pitch motor | 82 | 2 | 2 |
| | Attitude sensor | 210 | 5 | 4 |
| | Sum | 414 | 10 | 8 |
| Controller | Active (sampling, vehicle ctrl., etc.) | 1246 | 31 | 23 |
| | Sleeping | 782 | 20 | 14 |
| | Sum | 2028 | 51 | 37 |
| Sensors | Temperature and conductivity | 149 | 4 | 3 |
| | Depth (+ analog circuits) | 172 | 4 | 3 |
| | Sum | 321 | 8 | 6 |
| Telemetry | GPS and Iridium | 1014 | 26 | 19 |
| Total | | 5410 | 136 | 100 |

The controller (processor) is the most power-hungry main component with 37 % of the total energy expenditure (Table 1). This, however, is not because of complex control, but rather due to the fact that the processor of the glider is severely outdated. The controller of both Seagliders and Slocums is based on a processor design from the 1980s (the Motorola 68000-series) in a 1990s package (the Persistor). We estimate that the power consumption could be reduced by a factor of 4 for a modern processor based on a conservative application of Moore's Law.

Only 6 % of the total energy was expended on the CTD sensor – a figure that is arguably too low. We would like to allocate the savings from a new controller to sampling. Then the number of CTD samples could be increased and an $O_2$ optode (0.7 J sample$^{-1}$) could be included.

In this paper, we are mainly concerned with the energy expended by the buoyancy engine (Eq.(1) and Eq.(5)). Nevertheless, we allow for an additional 1 kJ per 2000 m dive to be allotted to vehicle heading and attitude control. This is justified by the fact that only 414 J were expended on this during the example 1000 m dive.

[The paragraph below should be read in the context of the second to last paragraph of the previous section 2.5 and the footnote there which states the specific energy content of lithium primary batteries].

Power budgets will be related to the vehicle volume as the displacement must make up for the weight of batteries. If we allocate 1/16$^{th}$ of a Watt (63 mW) to vehicle propulsion and heading control and another 1/16$^{th}$ of a Watt for the controller, sensors and telemetry, that would correspond to a 6.2 kg lithium battery pack for a two-year mission. Although challenging, it is possible to fit this battery into a vehicle with a displacement of 25 L. Please note how the example dive just falls slightly short of achieving the goal of 2/16$^{th}$ of a Watt (125 mW).

**2.8 Mission cost**

As a basis for estimating the mission cost we use the current costs for a core Argo float mission. The cost for the float itself is about 20 kUSD which approximately doubles when program management costs are included (Argo, 2019). Basing the cost estimate on Argo float costs can be justified for two reasons. The economy of scale for $O$(1000) slow gliders would approach that of floats rather than present gliders, and a winged float has many parts in common with regular floats; the hull, the buoyancy engine, GPS, Iridium, CTD, etc.

In Table 2 we include the additional costs for various glider specific items. A glider is inherently a more complex instrument than just a float with wings plus other components, and we also allow for costs associated with the increase in complexity of integrating the additional parts. Furthermore, we include a healthy profit of 50 % and development costs.

**Table 2.** Cost estimate for a slow glider mission based on an Argo float costs and Argo program costs.

| Item | cost (kUSD) |
|---|---|
| Core Argo float | 20 |
| Wings and fins | 1 |
| Roll and pitch assy | 5 |
| Attitude sensor and altimeter | 3 |
| Lager batteries | 3 |
| Complexity of integration | 10 |
| Profit of 50% on above | 21 |
| Amortization of dev. costs | 10 |
| Vehicle price | 73 |
| | |
| Argo program and data mgmt. | 20 |
| Mgmt. of complex program and data | 10 |
| Piloting (semi-automatic) | 10 |
| Launch | 5 |
| Recovery | 10 |
| Value of recovered vehicle | -10 |
| Program cost | 45 |
| | |
| Mission cost | 118 |

The simple budget in Table 2 indicates that a slow glider (winged float) mission would cost about 3 times more than an Argo float mission (40 kUSD). This may or may not be deemed prohibitive depending on scientific potential and value of such an endeavour.

**References**

Argo project web pages, FAQ – How much does the project cost and who pays?
http://www.argo.ucsd.edu/FAQ.html#cost last visited 21th of October 2019

**Response to Referee 1**

We apologize for the delay of our full response, but we opted to wait for the input from the second reviewer.

We thank the reviewer for the mindful comments on our manuscript. First, we want to comment the reviewer's general remarks below before proceeding to the detailed remarks. The referee's comment is given in *italic* font and blue colour followed by our response in regular font.

> *"The energy consumption by the sensors and controllers is marked as "beyond the scope of the research". I think that this is in fact a very important aspect."*

Referee number 2 (R#2) also points out this shortcoming of the paper, and we now address this aspect also, and focus not only on the energy consumed for propulsion. We now include a complete power breakdown of a low power glider dive (see Section 2.7 above).

> *"You do state that based on experience with a Seaglider, a cycle takes about 1 kJ for the electronics, which then would translate to 1/16 W, and could make the concept feasible."*

The 1 kJ we state is only for heading/attitude control per 2000m dive (this is included in our energy estimates). The detailed power analysis shows that this number is reasonable.

> *"… Slocum gliders, and, although the manufacturer has done a lot to reduce the power consumption of the electronics, a figure of 1-2 W is more appropriate. So clearly, sacrifices need to be made in terms what, how and how much is measured. "*

Indeed, but the Slocum glider focuses on sampling capabilities rather than ultra-low power operation. The Slocum science processor runs once every second potentially taking a sample every second. What if this was reduced to running the processor and sampling every 16 seconds? As far as the vertical resolution is concerned, we consider low sampling rates at a low vertical velocity to give adequate resolution for general hydrographic missions.

As R#1 points out later the controller must sleep most of the time. The Slocum glider has two processors and cycles them almost continuously. This is controlled by two master-data settings/sensors in the software, namely u_cycle_time(sec) and u_sci_cycle_time(sec). The cycle time might be increased for endurance missions.

> *"stratification may cause significant changes in the effective buoyancy drive, and as a consequence may require frequent monitoring of diving or climbing rates."*

True, but we believe this control problem is addressed for floats which also aim for low power and low buoyancy operation. The monitoring of depth rates should not have to be more frequent than regular sampling of the depth sensor.

We respond to this in the revised paper by adding the following to section 2.4: "The low excess buoyancy of 25 cc will be challenging to maintain over the dive in face of ocean in-situ stratification. We have stated the energy consumed to maintain this excess buoyancy as a continuous function in Eq.(1). The result of the calculation is depicted in Figure 2 (last panel) as a continuous curve. A real vertical velocity / buoyancy controller will discretise this curve as needed depending on the observed depth rate which might have to be monitored frequently."

*"…then simply reducing the buoyancy drive would allow for existing gliders to be used as described in the manuscript."*

We propose in the conclusions that: "Future work should firstly attempt to verify the concepts and findings presented here using existing gliders in the real ocean." If so, then one would be able to test the low buoyancy operation and potential upgrades to the vertical velocity control algorithm. We do not expect results that will render our concept infeasible.

*"… if it is at all possible to build a vehicle half the size that contains all the hardware needed to function and sufficient amount of batteries. If batteries is the limiting factor, a bigger glider may be more advantageous."*

We agree that present gliders indeed look compact and crammed enough already on the inside. Yet, Eq. (5) clearly shows that volume drives energy consumption. As energy considerations are of prime importance, vehicle volume must come down. This is achievable if the glider was designed with this consideration in mind from the start. This direction of development is necessary on the grounds of basic energy considerations. An example of a low volume vehicle is the SOLO-II float which has a volume of approximately 18 L – in its previous technological iteration, the SOLO-I float, it had a volume of 30 L (see table on second slide of Owens et al., 2012). Reduction of volume seems possible.

What if glider volume could only be reduced to 30 L one might ask and not the 25 L we call for. In this respect our paper is self-contained, since Eq. (5) is almost linear in volume and volume and energy consumption would both increase by roughly 20 %. This, we believe, would not invalidate the concept we propose.

In the revised version, we comment on the low volume challenge and integrate the above answer in section 2.5.

*"The final issue I have, is the costs of such a down sized glider. I reckon that a guide price of today's conventional glider is about $ 200k. To be deployed in thousands, the price must come down enormously."*

This issue is also raised by Referee 2 and we discuss it further in our paper. We now provide a cost estimate for slow glider missions (see above, Section 2.8, Table 2). This entails some uncertainties but is reasonably well justified and might be of interest to the reader.

*"Sections 4.1-4.3 show the results of such a glider that is deployed in various parts of the world's oceans. Personally I felt that each case conveys more or less the same message, and the one case would be as good as any other."*

We intended these missions to convey the same message in the sense that the proposed method of navigation (Eulerian roaming) is applicable in a pole-to-pole fashion in various scenarios. The real difference is how they supplement Argo-floats. In the Nordic Seas the slow glider is able to sample boundary currents, fronts and eddies in a manner that floats cannot do even if float density is high in the area. In the Gulf Stream and NAC mission we demonstrate how the slow glider may sample an intensified boundary current and the ensuing intense eddy field. Float coverage here is good but too low considering the energetic dynamics of the area. The slow glider will provide local snapshots of this variability. Finally, in the Drake Passage mission we demonstrate a mission in an area which is under-sampled by floats.

> *"Like one float, one glider would not tell much about the state of the ocean, and the appeal is a large number of devices. I thought that focussing on one case, where you look at how the added information of a Eulerian-roaming device, as opposed to a Lagrangian device would give, would be more compelling."*

Admittedly, what we would like to do in future work is to simulate how 100 slow gliders in the Gulf Stream and NAC would add significant (or not) scientific value to the floats in the area. The current paper is a necessary steppingstone introducing a novel concept with the intent to pursue such future work.

Below we respond to R#1's detailed remarks/comments.

> *(P1, L23): ""in that sense fall short of realizing his vision", this sentence suggests that as long as it has wings, all is well. I think what you mean is that the dynamic positioning is what is failing."*

We have changed this sentence to read: "and in this sense fall short of realizing his vision as far as wings also allow for dynamic positioning of the robots".

> *(P2, L4): ""simpler", simpler than what? Also related to this paragraph is that "just adding wings to a float" in reality comes with a serious increase in the level of complexity.*

We will omit the word simpler. The sentence will then read: "Floats, without wings, are now a robust and mature technology that has been developed since the 1950's …"

Note that we already in the next sentence state that gliders are "more complex [than floats]". We do not think that we should elaborate on the specific complexities, mainly heading/attitude control, in the introduction.

> *(P3, L16): "This sentence initially confused me, but it made sense after I looked up some details of the Argo float. I think the words "pause" and "parking" in this context are not clear for someone who is not very familiar with how floats are typically operated."*

We appreciate the referee's efforts to make sense of this sentence. To clarify this we will include a reference to http://www.argo.ucsd.edu/How_Argo_floats.html (Argo, 2019) at the end of the sentence. In the introduction we give ample float references and find we cannot elaborate further on float operations in the paper.

> *(P6, L9): "A considerable part of the lift is generated by the hull of the glider."*

> *(P7, L14-16): "(Related to the previous point) "… to compensate for …. smaller hull": this suggest that, at least for the Slocum gliders, the design of the wings (size/shape) is somehow optimized. I suspect it is not, as the leading principle in the design of the Slocum glider is easy construction and I don't think much thought has gone into the size of the wings."*

We acknowledge that the hull also contributes to lift. Probably, as R#1 suggests, not much thought has gone into the shape and size of the wings. For instance, Eq. (7) of Merckelbach et al. (2010) suggests that lift from the wings can be increased by 25% if the large sweep angle is reduced from 43 degrees to a more reasonable 10 degrees. If also made a little bit bigger, the wings should more than make up for the decrease in lift from a smaller hull.

In the paper we do not give a full account on the generation of lift. The main reason is that we are primarily concerned with the energy consumed, which is determined by drag.

> *(P8, L15): "Here I thought it might be difficult to achieve gliding with just 25 g of buoyancy, because the effective buoyancy may change more than this in a stratified ocean, and using a vehicle that has not a compressibility that is exactly matching that of seawater. This requires frequently adjusting the buoyancy on both the down and up casts. In my experience gliders typically reduce speed on both the down and up casts, due to stratification effects."*

It is true that the buoyancy control will have to be finer grained than what is currently implemented in gliders. Present gliders are not designed with low buoyancy operation in mind. However, floats manage to cope with this low-buoyancy and low-power control problem with neither excessive complexity to the controller nor excessive power consumption.

> *"You could, I suppose, store energy when reducing the volume on the down cast and releasing it again on the upcast, but still you will look at a hysteresis-like effect, and a much more technically complicated design (read: losing volume for batteries, increased costs)."*

R#1 here points to an interesting development of a self-regulating and recuperating buoyancy engine. We do not assume such a development in the paper and only presuppose regular buoyancy engines. Consequently, we will not have a more complicated design which adds vehicle volume and increases costs.

> *(P11, L19): "Here you say you set the glider's velocity vector. It is not clear to me where you specified the just that, the speed, or that you would specify the buoyancy of 25 cc. I guess you prescribed the speed."*

R#1 is correct that we prescribe the speed given that we in Section 2.4 (Figure 3) establish the operating point for the buoyancy to be 25 cc (for a speed of 13 cm s$^{-1}$ in the horizontal plane). We have clarified this by adding the following sentence to the paragraph:

"The speed of 13 cm s$^{-1}$ in the horizontal plane was established in section 2.4 (Figure 3) for the operating point of 25 cc in excess buoyancy."

> *"In that case, my previous point should some how be addressed. If you specified the buoyancy, I suggest you include a small discussion on how frequently the buoyancy needs to be changed, and what the energetic costs are."*

We do include these costs in energy to maintain an excess buoyancy of 25 cc as we evaluate energy consumption using Eq.(1) in conjunction with the salinity and temperature fields from the reanalysis product. Such a calculation is exemplified in Figure 2. The buoyancy engine must pump where the energy needs to be increased (last panel of Figure 2). We evaluate the continuous integral of Eq.(1) but do not suggest how this should be discretized by a real vertical velocity controller.

Such a vertical velocity controller would not have to be significantly different than what is currently implemented in floats and gliders. The Seaglider, for instance, monitors the pressure rate and pumps if depth rate sinks below a certain value (i.e. simple threshold control). We do not see why we would need a substantially more complex and more energy-consuming controller.

*(P14, L12): "Here (and also in following paragraphs) you specify the energy consumption. I read this number as 691 cycles, at 1 cycle a day, using 1/16 W gives 2.9 MJ. Or is this computed using equation 5, taking into account the actual velocity the glider made, and stratification it faced, and the effort done to compensate for it? Also I think, this includes only the power required for propulsion. So what about the electronics?"*

R#1 is confused and rightly so. We were not clear on how we calculate the energy consumption and have expanded this paragraph as follows:

"The glider performed 691 cycles and the energy consumption was 2.9 MJ (or 2.3 kg of Lithium primary batteries). This is calculated by evaluating Eq.(1) using the established operating point with an excess buoyancy of 25 cc and using the salinity and temperature fields of the reanalysis product. Then 1 kJ is added per dive for heading/attitude control and finally 0.5 kJ is added for surface pumping to raise the antenna out of the water. The full EOS of water and hull (Eq.(2)) is taken into consideration. Values for compressibility and thermal expansion are as given in Section 2.1 and the result of the calculation is depicted in Figure 2 panel d)."

As far as the electronics is concerned, it is hard to account for it given that the main component of the power consumption is an obsolete processor (see Section 2.7 and the low power dive exemplified there). Notice how we do include energy consumed by the electro-mechanics needed for vehicle heading and attitude control and the associated sensor.

**New References**

Owens B., Roemmich D. and Dufour J.: Status of SOLO-II Floats Development, presentation given to the Argo Steering Team meeting No. 13, Paris, France, 2012 http://www.argo.ucsd.edu/AST13_SOLO-II_Status.pdf

Argo web pages, How do Argo floats work, http://www.argo.ucsd.edu/How_Argo_floats.html last visited 18th of October 2019

**Response to Referee 2**

Thank you for your thoughtful comments on our manuscript. Your general comments suggest that we solve the following generalized ocean observing problem:

> Given $n_f$ floats, $n_{rg}$ regular gliders and $D_a$ altimetric data
> Show that $n_{sg}$ slow gliders would add scientific value in an economic fashion

This is indeed an interesting and necessary problem for an Observing Systems Simulation Experiment (OSSE) which must be undertaken before the oceanographic community implements Stommel's vision. Such an OSSE, we argue, is beyond the scope of our manuscript and present study. Note that this manuscript forms a necessary steppingstone for such an OSSE and motivates further research.

On several occasions concerns over the costs of such slow glider missions have been expressed, and we have addressed this better in the revised version of our paper. Please see our letter to the editor and the referees (Section 2.8 above).

We have added the following paragraph to our Section 4.6 to alleviate your concern that this undertaking will become a waste of resources better directed at other more established methods:

"While the tracks of the slow glider (/winged float) presented in this section clearly demonstrate oceanographic *potential* it remains to prove scientific *value* added to the existing network of Argo floats, regular gliders and altimetry. The scientific value could be explored and possibly quantified by an Observing System Simulation Experiment which would include all observing elements of the GOOS including our slow virtual glider. Such future work might build on the concepts and methods presented here. In Section 2.8 we roughly estimate that such glider missions will cost 3 times more than a float mission, which requires that the scientific value be correspondingly enhanced if Stommel's vision is to be implemented in the form of slow gliders as we propose."

Below we provide answers to the referee's comments which are given in *italic* font and blue colour followed by our response in regular font.

> *"Attempts to justify the Eulerian roaming were not convincing, especially for single missions."*

As far as the navigation in areas of key oceanographic interest is concerned, we disagree. The single missions demonstrate clear scientific potential using the Eulerian roaming approach in various scenarios. Notice how the slow glider employing Eulerian roaming successfully and in an oceanographic meaningful way navigates an important marginal sea, an intensified boundary current and eddy field, and the remote Southern Ocean. However, we do not attempt to explore how a fleet of roaming gliders would generate scientific value. This relates to our comments above.

> *"What scientific questions beyond the lucky detection of an episode or feature would such a large scale network address that ARGO floats do not already address?"*

We consistently target high-value oceanographic features. In the Nordic Seas, for instance, we target major oceanographic features and pathways. This includes fronts, boundary currents and eddies. Most of these go un- or under-sampled by the Argo floats who rely strictly on Lagrangian luck.

> *"What is so special about 1000 gliders exactly? Are the reasons Stommel used to justify that number still relevant today?"*

Stommel meant $O$(1000) gliders and never gave a scientific rationale for the number, instead presenting his vision as a short science fiction story. The 1000 gliders was a number he mentioned probably based on his intuition and gut feeling. But as far as oceanographic intuition and gut feelings go, Stommel's cannot be easily dismissed. Hence, we start from his order of magnitude as a reference. The design specification of the Argo array mentions the potential of a gliding float with wings. Taken together, we suggest that these two sources form a relevant premise for the paper.

> *"Central to this question is cost, which was essentially brushed aside.. Presently, gliders cost about 10 times more than floats to purchase."*

True, and we have addressed the costs in a separate section (2.8). We then argue that the price of the slow glider is a factor of 3 cheaper than present gliders given $O$(1000) units.

> *(P7, L10) : "it may be stated that those glider manufacturers now have different designs and that performance may differ (e.g. Seaglider ogive fairing or larger Slocum G3 hull). It would be interesting to update the results for those and to run more simulations for reduced volume versions, rather than just one."*

We are not aware of any hydrodynamic models and associated coefficients for the Slocum G3 or the Seaglider Ogive fairing. We believe we must stick to established models in this section. Also, please note how little is gained by a glider with 20 % reduced drag in terms of velocity. Simulating more drag coefficients would clutter Figure 3 with clusters of intersecting lines making the figure harder to read. We ask that we may keep the figure as is.

> *(P9, L20-25): "It is not clear if a CTD-only glider will best serve the global observing system: there are many more Essential Ocean Variables that gliders can (and soon will) be able to measure. This flexibility is one of the strengths of current gliders. Some examination of what payloads would be possible compared to what is normally done now would be interesting, and I think not outside the scope of the paper. Later in the paper, microstructure is mentioned. That paragraph could be expanded to include other potential payloads for the small glider."*

In addition to the microstructure we also mention the Argo biogeochemical (BGC) suite. However, we must admit that we cannot fit these into our small ultra-low power glider – neither volume wise nor power wise. At present, we would like to add. However, three future developments are likely to improve the situation. Batteries will have larger capacities, and sensors will become smaller and consume less power thus making a small slow BGC glider possible. A slow glider would also provide a depth averaged current which is more useful than the 1000 db (typical) current produced by floats.

The concept we propose consists of three elements; smallness, slowness and a novel way of navigation (Eulerian roaming). This novel concept needs not be taken wholesale. Existing gliders using existing sensor suites measuring more EOVs may be operated according to the principles of

slowness and Eulerian roaming. In particular, the slow speed may conserve a lot of power (Eq. (3)) which may be directed at powering advanced EOV sensor suites.

*"I am not sure why detailed power budgets and engineering calculations should be excluded from the paper. It seems to me that would strongly support the main point of the paper."*

Agreed. R#1 also laments this short-coming, and we have thus decided that we need to provide a complete power budget for a low power Seaglider dive. Please see the beginning of our letter to reviewers and the editor for a new section (2.7) on the matter.

*"More details about the strengths and weaknesses of Eulerian roaming are necessary if the reader is to believe this is a viable alternative. The simulations following help, but no indications are given on how such data could be/have been handled other than a simple citation (Todd et al., 2016)."*

Agreed. Todd et al show that even highly irregular glider tracks obtained using "current-crossing navigation" (similar to our Eulerian roaming) in energetic regions (Todd et al., 2016, figure 1) can be used to obtain valuable oceanographic measurements. This is supportive of the insight that can be gained from Eulerian roaming; however, caution is needed when calculation distribution of geostrophic currents and related parameters. We now expand on the concept, assumptions, and also reference a recent paper. The paragraph at (P20, L3) will be augmented with an additional paragraph as follows:

"The key assumption in using the local streamwise coordinate system for geostrophic current calculations along the glider trajectory is that all flow is parallel to the depth-averaged current (DAC). When the depth-average current direction is not perpendicular to the transect segment of the glider path, a decomposition into cross-track and along-track components must be made. In these conditions, using the currents from the local streamwise coordinate system will be in error; however, the transport will remain relatively unaffected.  In a recent study, Bosse and Fer (2019) reported geostrophic velocities associated with the Norwegian Atlantic Front Current along the Mohn Ridge, using Seaglider data, following Todd et al. (2016) and assuming DAC is aligned with the baroclinic surface jet. They also calculated the geostrophic velocities and transports using the traditional method, i.e. across a glider track line, and found that the peak velocities of the frontal jet were 10-20 % smaller but the volume transports were identical to within error estimates. The Eulerian roaming can thus be used to obtain representative volume transport estimates of relatively well-defined currents. We also note that the present 1000-m depth capability of gliders limits our ability to compose the geostrophic currents into barotropic and baroclinic components in water depths substantially deeper than 1000 m. A 2000-m range will allow us to more reliably approximate barotropic currents as the depth averaged profile, up to depths of around 2000 m."

*"This section 2.6 seems out of place, and fits better in the next section."*

We agree and have moved section 2.6 to 3.1.

*"Section 4. Results and Discussion. The hypothetical case studies are interesting and show the potential, but are not convincing in terms of scientific value. An attempt is made in 4.4, but the analysis from the mission is oversimplified in my opinion. Separating temporal and spatial variability on these year long missions over large horizontal gradients would be very*

*difficult and it is not always possible with one long track to collect data "useful in understanding the role" or that will "capture the properties and variability". The section about altimetry begins to touch on what could be the scientific goal of such a fleet: the surface topography problem. The number of gliders needed to reduce the current errors in the altimetric eddy field (number, phase and intensity) could be quantified in this paper and justify the existence of the fleet."*

We agree with the reviewer that separating temporal and spatial variability requires a fleet of gliders. It was not our intention to claim that a single glider would capture the properties and variability of different current systems. We clearly state, in the end of the section: "While we have shown tracks of individual gliders, it should be clear that the impact of a slow roaming glider concept will increase when employed in large numbers. Also, the simulations here where a few gliders are hand-piloted does not show the full potential of the approach." We exemplify the potential skill of single missions, and advocate for deployment in numbers. Authors, reviewers and the reader all know that only then we can harvest meaningful information from the missions and describe the properties and variability of the ocean circulation and dynamics. We close this response by repeating that a demonstration of the full potential of the approach with simulations of numerous missions is left for future research. Careful considerations and optimization must be made to design a suite of missions. Even a naïve approach of deploying, from the same location and with similar target missions, every month for a duration of 1 year (12 deployments) would return a highly informative data set and would allow sufficient averaging and separation of temporal variability. Exchanges across the Greenland and Norwegian Seas are poorly known, transport of the frontal branch of the Atlantic Water in the Norwegian Sea is poorly known, the return Atlantic current in Fram Strait is poorly known, the role of winter convection in water mass transformations in the Nordic Seas is poorly known. While targeted, regular glider missions would help filling gaps in our knowledge, Eulerian roaming of a fleet of slow gliders would be complementary and provide a different mapping capability between Argo floats and regular gliders.

*"Section 5. Specific methods of piloting large numbers should be cited (optimal fleet mission planning) as well as the scientific objectives one might achieve with this (e.g. optimized for data assimilation for altimetry or some other objective). This was very briefly touched upon in the conclusions and future work, but really this should provide a solid background to why the reader should even dig into the paper. Clearly this concept is most valuable in a complex large fleet sampling context and some work has been done already."*

The reference, L'Hévéder et al., 2013, we provide in the conclusion and further work is the most relevant citation and appropriate starting point for continued research. In their work, which is centred around an OSSE, they study how a fleet of gliders could reconstruct a mesoscale temperature field. We believe the conclusion (Section 5) should be as succinct as possible and would like to keep the conclusion as is. However, R#2 correctly points out that we need to better address the problem of fleet control with respect to a certain scientific objective. Notice that this is a vast topic (Rudnick, 2016) and that we will only be able to scratch the surface. Still, we would like to add the following paragraph to our discussion (Section 4.4):

"The control and steering of a network or fleet of slow gliders should aim to optimize for some scientific objective possibly in conjunction with other sensing platforms. Alvarez and co-workers (2007) have looked at synergies between floats and gliders to improve reconstruction of the

temperature field. Synergies also exists between a glider fleet and altimetry to map geostrophic currents (Alvarez et al., 2013). We suggest that future work should see the slow glider concept not as a homogenous fleet, but rather as a part of a heterogenous suite of ocean sensing technologies. The topology of the network needs some consideration and one interesting option is to cluster the gliders in and near an oceanographic feature to explore it in greater detail. Some in-situ experiments with glider fleets have been conducted (e.g. Leonard et al., 2010; Lermusiaux et al., 2017a). The problem of planning optimal paths for gliders is reviewed by Lermusiaux et al. (2017b)."

**New references**

Alvarez A., Chiggiato J. and Schroeder K.: Mapping sub-surface geostrophic currents from altimetry and a fleet of gliders, Deep Sea Research Part I, 74, 115-129, doi: https://doi.org/10.1016/j.dsr.2012.10.014, 2013

Alvarez A., Garau B. and Caiti A.: Combining networks of drifting profiling floats and gliders for adaptive sampling of the Ocean, IEEE International Conference on Robotics and Automation, Roma, Italy, 10-14 April, doi: https://doi.org/10.1109/ROBOT.2007.363780, 2007

Bosse, A. and Fer, I.: Mean structure and seasonality of the Norwegian Atlantic Front Current along the Mohn Ridge from repeated glider transects, Geophysical Research Letters, 46, doi: https://doi.org/10.1029/2019GL084723, 2019

Leonard, N. E., Paley D. A., Davis R. E., Fratantoni D. M., Lekien F. and Zhang F.: Coordinated control of an underwater glider fleet in an adaptive ocean sampling field experiment in Monterey Bay, J. Field Robot., 27, 718–740, doi: https://doi.org/10.1002/rob.20366, 2010

Lermusiaux P.F.J., Haley Jr. P.J., Jana S., Gupta A., Kulkarni C.S., Mirabito C., Ali W.H., Subramani D.N., Dutt A., Lin J., Shcherbina A.Y., Lee C.M. and Gangopadhyay A.: Optimal planning and sampling predictions for autonomous and Lagrangian platforms and sensors in the northern Arabian Sea, Oceanography 30, 172–185, doi: https://doi.org/10.5670/oceanog.2017.242, 2017a

Lermusiaux P.F.J., Subramani D.N., Lin J., Kulkarni C.S., Gupta A., Dutt A., Lolla T., Haley Jr. P.J., Ali W.H., Mirabito C. and Jana S.: A future for intelligent autonomous ocean observing systems, Journal of Marine Research, 75, 765–813, doi: https://doi.org/10.1357/002224017823524035, 2017b

---

## Author Comment (AC3) · 4 Nov 2019

Dear Referee 1,

please refer to our author comment (AC2) below for a full response to your review of our manuscript.

We apologize for the inconvenience, but we opted to provide one comprehensive document. Apparently the OSD web pages has it that we must provide this as a response to one of the reviewers.

Kind regards, E.M. Bruvik
* * *

---

## Author Response (AR2)

Dear Referee #1,

We thank you for your work on our revised manuscript. Below we briefly answer your minor remarks / requests for the technical corrections.

*(p.2/l.13) The use of "Only" gives this sentence an unnecessary negative sound. Perhaps you can consider an alternative (However, or replace only by though used later in the sentence).*

We agree and removed "only".

*reread the larger parts of inserted text in the revised manuscript, as I think they don't connect always so well with the context they are placed in. Reshuffle or provide a connecting sentence.*

We have looked at the text as a whole again and do not find room for significant improvements without major rewrites. We must also at this stage be careful not to induce changes of meaning as the text is largely approved by R#2. We have however attempted minor edits and rephrasing throughout the text which improved the connectivity in the content.

*Also I would like to ask you to carefully check the references. I noticed that McMasters and Schmitz are not listed in the references. Perhaps others are missing too. I didn't check all of them.*

Thanks for noticing this oversight on our behalf. We cross-checked and listed all references.

*p6 l29 Recently work by Merckelbach and coworkers was published building upon the gliderflight model you reference (Merckelbach 2010). In the recent publication the equations are correct and the model has been validated using data from a glider mounted Profiling Doppler Velocity Log, and a microrider mounted Electromagnetic current meter. You may refer to this paper instead (A Dynamic Flight Model for Slocum Gliders and Implications for Turbulence Microstructure Measurements Merckelbach et al, Journal of Oceanic and Atmospheric Technology 2019, DOI: 10.1175/JTECH-D-18-0168.1)*

We do reference Merckelbach et al. (2019) as an example of a field test where the glider velocity is measured directly (p8,l9). However, R#1 is correct to point out that we need to reference this paper also in our list of available hydrodynamic models. We corrected this.

*p.10/l.8 You estimate that 1/16 W equals 5.4 kJ/day. That is expressing the same thing in different units. It is also a bit of a puzzle for the reader how to arrive at the numbers mentioned in this section.*

We express the same quantity in different units to aid the reader with understanding as it is otherwise not so clear what 1/16 W means in terms of practical energy consumption per time. We reworded as follows:

"Since 1/16 Watt corresponds to 5.4 kJ day-1, there will be 1.9 kJ remaining to expend on vehicle compressibility...".

*You make a remark on using a free-flush CTD to save energy, and continue arguing that it is technically possibly to correct for errors in CTD readings due to thermal lag effects*

*(p.9/l26). Then in Table 1 it is mentioned that 860 CTD samples are taken in about 11 hours, or 1 per minute. At this rate it is not possible to do any corrections at all. Even using the maximum sampling rate available on standard glider CTDs, namely 1 Hz, the correction is dubious when strong temperature gradients are present. Even if a 4x as efficient processor is used and all the energy is used to sample more frequent, then you would get at 1 sample every 5 seconds, which is not good enough for these corrections.*

We agree with R#1 regarding the considerations for sampling and possible corrections of the CTD. We amended:

"Only 6 % of the total energy was expended on the CTD sensor – a figure that should arguably be increased in order to apply appropriate corrections. We would like to allocate savings from a new controller to increasing the number of CTD samples and possibly including an $O_2$ optode (0.7 J sample$^{-1}$).".

On the particular mission from which the example (Table 1) is taken, we used non-uniform sampling and sampled the mixed layer with a sampling time of 20 s. This is now clarified in the table caption. On this mission we were forced to run an extremely low sampling rate given the endurance nature of the mission (9.5 months). This resulted in data of fair quality yielding valuable oceanographic measurements (Våge et al, 2018). We do not want to go further into salinity data quality discussions and believe that we have provided enough cautionary comments to the reader on this topic.

Further we now reword the following:

"For the vision presented here, power-hungry sensors must be avoided. It is doubtful whether a pumped C-T system could be employed on a slow glider. Unpumped conductivity cells have successfully been used in gliders, and after appropriate corrections (Lueck and Picklo, 1990; Garau et al., 2011) supply data of adequate quality. Such corrections will be challenging for a relatively slow flow past the sensor in a slow glider, but technically possible provided an adequate sampling rate and flushing of the conductivity cell (K. Martini, personal communication, 2019). Further calibrations and bias removal will also be possible against Argo floats and ship-based measurements. The user must carefully assess the accuracy needed for salinity against a trade off from endurance."

*Table2: I find the break-down of costs not very convincing. More than once I heard from people in industry that the selling price of such instruments is normally about 10x the construction cost. Would you call that profit/recovering development costs? Your numbers for profit seem very modest. On the other hand. then numbers you quote for components seem rather expensive. So I wonder where those numbers come from. If they are (educated) guesses I would be reluctant to specify them, although I agree with you that a price of about 70k would seem realistic, in particular if scaling arguments apply.*

We are content that R#1 finds our total price estimate of 73 kUSD in Table 2 to be realistic. We assume that a profit of 50% is acceptable to manufacturers and customers alike. We might be wrong though and potential manufacturers might demand or need a higher profit. On the other hand, as R#1 points out, our component costs might be on the high side. These costs are educated guesses based on our experience with buying spare/repair components for our gliders. Specialists with more know-how on the glider costs could scale or re-distribute our cost estimates as necessary. We would

like to inquire glider manufacturers on the profit margin but expect such information to be a business secret. We are thus forced to make some guess-work in putting together the simple budget of Table 2. At this stage, we retain the Table 2 as is. In text we inserted:

"While the relative distribution of profit, component costs and operation costs can be different, the overall cost estimate is deemed representative. "

Dear Referee #2,

We appreciate your input on our revised paper. Below we address your suggestions for minor revisions.

*2.1 a quick comment on why the red and blue lines in Fig. 2 are different may be useful.*

We state (in the text, but not in the figure caption) that: "Equation (1) approximates the energy consumption well, but slightly overestimates compared to the full EOS formulation. This is primarily due to Eq. (1) neglecting the thermal expansion of the hull (which will assist the vehicle in reaching the surface)." To further clarify, we replaced "This is primarily" with "The difference (compare blue and red lines in Fig 2d) is primarily…"

*2.2 it seems a bit confusing to have a reference total velocity in Eq. 4 as 13 cm/s, but then use that regularly as a horizontal velocity. Is the effect simply too small to worry about in the power calculations?*

We use Eq. (4), relating power to velocity, only to support the general and rough claim that given the present maxim of "1/2 knot at 1/2W" one may rather choose to operate at "1/4 knot at 1/16W". This is roughly speaking and at gentler glide slopes the velocity through water will be close to the horizontal speed. Also Eq.(4) is supportive of the insight that velocity is a costly affair in terms of power. All of our detailed energy calculations and considerations are based on Eq.(1) and Eq.(5).

*2.4 any comment on the reasons for the difference between SG and SL in Fig. 3? Is it basically a larger volume of SG, hence more skin friction compared to SL? Why would this change at higher net buoyancy? (a curiosity question, may not be needed in text).*

In Figure 3 the velocity polars of the Seaglider and the Slocum are plotted as calculated from the cited hydrodynamic models. As R#2 points out the Seaglider has a relatively larger surface area hence more drag. At higher velocities and buoyancies, however, the laminar flow profile of the Seaglider begins to shine thus resulting in a higher velocity vis-à-vis the Slocum glider. To clarify, we inserted:

"The Seaglider has a relatively larger surface area hence more drag. At higher velocities and buoyancies, however, the laminar flow profile of the Seaglider improves the performance relative to the Slocum glider."

*-what about the difficulties for 2000 m rated hulls? Won't this imply thicker Al walls and hence less room for internal electronics and batteries? Especially if a compressible design such as SG is used. Is 25-30L still a reasonable target? Larger hull mass also implies more displacement and overall volume needed.*

R#2 correctly points out the challenges presented by a 2000 m rated hull. We inserted the following paragraph to the end of Section 2.5:

"The 2000 m hull of the vehicle must satisfy three requirements. It must be strong enough to withstand pressure, yet the compressibility should match that of sea water, and finally, offer the necessary payload volume for batteries, electronics and the buoyancy engine. This poses a real engineering challenge. Jenkins et al (2003, Sect. 6.3) contains detailed considerations for an aluminum hull. However, it is likely that alternative composite materials must be considered for the hull (Osse and Eriksen, 2007; Webb, 2005)."

*2.7 Table 2 is useful. However, a little confusing because Argo is mentioned in several places when in fact you are estimating the slow glider cost (with Argo as a guide, if I understand the text correctly). Also, maybe I missed it, but shouldn't the cost in Table 2 is for the first mission of the slow glider only. Wouldn't the 2nd, 3rd, and so on be significantly less? You could value the returned glider at higher value than the -10k cost, for example. This would quickly make a more competitive case with Argo. (also from environmental standpoint as you mention)*

R#2's understanding is correct as far as we use Argo float cost and Argo mission cost as a basis for estimating slow glider (/winged float) mission cost (vehicle price + program cost). Note however that we distribute development costs evenly out on the missions, hence the constant overall mission cost. The value of the recovered vehicle is set at a moderate level to make the budget not overly optimistic. We might note that there is also an opportunity for post mission calibrations if the vehicle is recovered.

*4.1 "eastward to Greenland" should be "westward" I think.*

Corrected..

*4.4 "targets to observing" is awkward...try "targets the observation of". "highly needed and" should be removed.*

Changed as recommended.

*4.5 "launched off the coast..." replace with "launched and recovered near the coast"*

Done.

*5. Conclusions: reword "expectations to" as "expectations for"*

Done.

[revised manuscript text omitted]